# EvoMAS: Heuristics in the Loop—Evolving Smarter Agentic Workflows

## Abstract

The rapid development of Large Language Models has driven Multi-Agent Systems (MAS) growth, but constructing efficient MAS requires labor-intensive manual design. Current automation methods generate templated agents, use monolithic optimization, and ignore task complexity gradients. This paper presents Evolutionary MAS (**EvoMAS**), a biologically-inspired framework that systematically addresses these limitations through three interconnected dimensions: (1) **dynamic and diverse evolutionary strategies** with six biologically-inspired operators (3 exploration, 3 exploitation) and adaptive strategy selection; (2) **role-level evolution** that dynamically optimizes agent specialization and collaboration patterns; and (3) **curriculum-guided evolution** partitioning tasks by difficulty levels and evolving sequentially from simple to complex with cross-stage stability constraints. Additionally, to resolve the contradiction between the inefficiency of pure evolutionary methods and the limited flexibility of manual design, we developed the **"Cyber Creator"**, a meta-control system combining dynamic rule formulation with reflective updates. Experimental evaluations demonstrate that EvoMAS consistently outperforms existing methods across multiple domains while maintaining cost efficiency, with agent roles dynamically evolving from homogeneous actors to specialized reasoning ensembles. Codes are available at EvoMAS.

## 1 Introduction

The rapid advancement of Large Language Models (LLMs), particularly the flourishing ecosystem of Model Context Protocol (MCP) (Hou et al., 2025), has propelled MAS as a powerful collaborative paradigm at the forefront of AI innovation (Li et al., 2024a; Han et al., 2024; Cemri et al., 2025). However, current MAS design methodologies face fundamental challenges: they predominantly rely on static predefined architectures and fixed interaction patterns—a rigid design philosophy that severely constrains their ability to respond to complex and dynamic environments. While such systems may excel in specific scenarios, they exhibit notable *adaptation barriers* when confronting open-ended, dynamic problems. Consequently, the automation and optimization of MAS design has emerged as a critical frontier challenge (Weyns & Oquendo, 2019).

Recent years have witnessed significant progress in agent system automation technologies, albeit with evident bifurcation trends. One category focuses on single-dimensional optimization: DsPy (Khattab et al., 2024a) and EvoPrompting (Chen et al., 2023a) pioneered automated paradigms in prompt engineering; GPTSwarm (Zhuge et al., 2024) and G-Designer (Zhang et al., 2024a) dedicated efforts to optimizing inter-agent communication protocols; while EvoAgent (Yuan et al., 2024) and AutoAgents (Chen et al., 2023b) explored the possibilities of single-agent self-evolution. Despite their respective strengths, these approaches struggle to achieve system-level breakthroughs due to their localized optimization perspectives. Another category, including ADAS (Hu et al., 2024), AgentSquare (Shang et al., 2024), and AFlow (Zhang et al., 2024b), attempts to expand the design search space, constructing optimized workflows on specific datasets through heuristic search, Monte Carlo tree search (MCTS), or evolutionary algorithms, demonstrating capabilities surpassing manually designed systems. Nevertheless, these methods reveal **severe limitations** when facing cross-domain tasks: ❶ they typically employ **singular, fixed optimization strategies** inadequate for diverse task requirements; ❷ their system structures and agent profiling remain excessively **template-based**, lacking necessary flexibility and innovation potential; ❸ they **disregard the crucial impact of task**

Figure 1: Motivation: Three Evolution Dimensions in EvoMAS. EvoMAS explores agentic workflow evolution from three key dimensions: (a) role-level evolution enhances agent specialization and coordination, (b) diverse strategies enable LLM-guided exploration with reflection-based updates, and (c) curriculum-guided evolution promotes gradual adaptation across tasks of increasing difficulty.

**difficulty gradients** on learning efficiency, resulting in insufficient generalization capabilities in complex scenarios.

Addressing these challenges, we introduce **EvoMAS** —a biologically-inspired framework for automated evolution of multi-agent systems. EvoMAS integrates three interconnected evolutionary dimensions as shown in Figure 1: ❶ **role dimension:** the system implements adaptive role evolution mechanisms that dynamically refine agent specializations and interaction patterns throughout the evolutionary process. This transcends traditional rigid agent roles, enabling specialized collaboration tailored to task requirements; ❷ **strategy dimension:** EvoMAS constructs a dual-track "exploration-exploitation" evolutionary mechanism encompassing six biologically-inspired strategies, enabling the system to achieve an exquisite balance between search space diversity and optimal solution convergence; ❸ **learning path dimension:** the system implements a curriculum-inspired progressive adaptation process, enabling complex capabilities to build gradually upon simpler tasks, significantly enhancing cross-task generalization abilities.

While natural selection provides powerful optimization principles, purely evolutionary approaches can be inefficient for complex tasks, and conversely, rigid human design often limits adaptability. To address this fundamental tension, EvoMAS introduces the **"Cyber Creator"**—a meta-control system that strategically combines rule-based guidance with adaptive learning. This mechanism bridges the gap between undirected

Table 1: *Comparative Analysis of Core Capabilities Across MAS Automation Frameworks.*

| Method | Multi-Agent Evol. | Dynamic Strategy | Meta-Evol. | Curric. Learn. |
|---|---|---|---|---|
| EvoFlow | ✔ | ✗ | ✗ | ✗ |
| FunSearch | ✗ | ✗ | ✗ | ✗ |
| EvoAgent | ✔ | ✗ | ✗ | ✗ |
| AFlow | ✔ | ✗ | ✗ | ✗ |
| ADAS | ✔ | ✗ | ✗ | ✗ |
| **EvoMAS** | ✔ | ✔ | ✔ | ✔ |

evolution and artificial intervention through explicit rule-setting and periodic reflective updates. Additionally, EvoMAS employs graph structures to precisely express the topological and functional characteristics of multi-agent workflows, while constructing an evolutionary resource center comprising rule pools and gene pools that provides robust support for knowledge accumulation and transfer. Table 1 illustrates how EvoMAS distinguishes itself from existing automated design methods through comprehensive evolutionary capabilities, being the only framework to simultaneously support dynamic strategy evolution, meta-level adaptation, and curriculum-guided learning. EvoMAS achieves state-of-the-art(SOTA) performance across six benchmarks, while maintaining superior cost-efficiency, outperforming both manual designs and automated baselines.

## 2 RELATED WORK

**Agentic Workflow.** With the increasing capabilities of LLMs, the paradigm of Agentic Workflow has emerged as a promising approach to construct structured and multi-stage task-solving processes Hong et al. (2024a); Zhang et al. (2024c); Wang et al. (2023a). This paradigm typically comprises multiple LLM-invoking nodes with well-defined inputs and outputs, organized in the form of graphs, code, or flowcharts to specify the execution sequence.

Existing research in this area can be broadly categorized into two directions: general-purpose workflows (Madaan et al., 2023; Wang et al., 2023b) and domain-specific pipelines (Zhong et al.,

2024; Xu et al., 2024). The former focuses on universal reasoning strategies, while the latter builds tailored structures for specific tasks such as code generation (Ridnik et al., 2024; Hong et al., 2024b), data analysis (Zhou et al., 2023; Ye et al., 2024), and multi-hop question answering (Zhou et al., 2024). However, most approaches rely on predefined templates or operator libraries and lack the expressiveness for hierarchical structural evolution and dynamic adaptation.

**Automated Agentic Optimization.** To alleviate the burden of manually designing complex workflows, research on automated agentic workflow optimization (Zhuge et al., 2024; Li et al., 2024b; Hu et al., 2024; Zhang et al., 2025b) is gaining traction. Some approaches focus on prompt (Khattab et al., 2024b; Fernando et al., 2024; Yang et al., 2024; Yüksekgönül et al., 2024) or parameter tuning Saad-Falcon et al. (2024), while others aim at optimizing the structural composition of workflows, including inter-module connectivity, execution ordering, and conditional dependencies.

Representative methods include ADAS (Hu et al., 2024), which linearizes workflow code and performs sequential structure search, and GPTSwarm (Zhuge et al., 2024), which models workflows as graphs and uses reinforcement learning for structural optimization. Additionally, AFlow (Zhang et al., 2024b) encodes workflows as code and applies MCTS to explore efficient execution paths, demonstrating superior performance over manual designs. However, these methods still suffer from limited search efficiency, constrained expressiveness, and poor cross-task generalization. In particular, they lack effective mechanisms for heterogeneous module collaboration and feedback-driven structural evolution, limiting their ability to adapt to complex multi-faceted tasks.

## 3 PRELIMINARY

This section establishes the theoretical foundation for automated multi-agent system design by introducing core design assumptions, formalizing the graph-based structural representation, and defining the constrained optimization problem that guides our evolutionary framework.

### 3.1 REPRESENTATION: GRAPH-BASED FORMULATION OF MAS

To capture the complex control flow, information exchange, and collaborative dynamics inherent in multi-agent systems, we model MAS workflows as sparse, cyclic directed graphs as illustrated in Fig. 2. Formally, a workflow is represented as $G = (V, E)$, where $V = \{v_1, v_2, \ldots, v_n\}$ denotes the vertex set with each node $v_i$ corresponding to an autonomous agent, and $E \subseteq V \times V$ represents the edge set encoding directed dependencies for information flow and control signal propagation. The graph maintains structural integrity through a unique source node (input) and sink node (output), ensuring well-defined task boundaries and deterministic execution semantics.

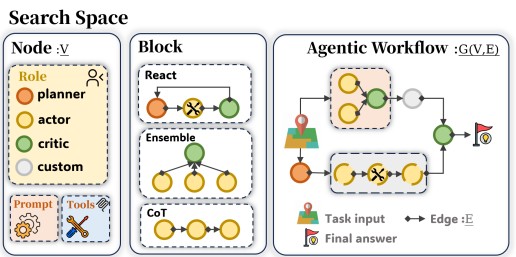

Figure 2: Agentic workflow search space represented as graphs with role-specific nodes and modular blocks, enabling dynamic tool use, prompt chaining, and structured multi-agent task solving.

### 3.2 DESIGN ASSUMPTIONS

Our approach to automated multi-agent system evolution is built upon three core assumptions that define the theoretical foundation and constrain the design space:

$\mathcal{H}_1$ (**Unity in Diversity**). *Following Ricardo's comparative advantage theory, specialized agents achieve collective efficiency even when individual capabilities differ*. We assume that heterogeneous agent roles with complementary specializations—some focusing on reasoning, others on verification—yield superior performance compared to homogeneous configurations (Bettini et al., 2023; Cao et al., 2019).

$\mathcal{H}_2$ (**Know Thyself**). *Based on Bayesian learning principles ($P(\theta|D) \propto P(D|\theta)P(\theta)$), agents continuously adapt through feedback loops*. We assume that self-monitoring mechanisms enable

Figure 3: Overall EvoMAS Framework. The framework follows a three-stage process: (1) initialization via RAG-based retrieval from a knowledge-rich resource library, (2) population evolution through LLM-based variation and selection, and (3) evolution resource library update based on the results.

robust error correction and strategic refinement, allowing systems to maintain stability while adapting to evolving requirements (Bilal et al., 2025).

$\mathcal{H}_3$ (**Less is More**). *Following Occam's Razor and bias-variance tradeoff principles, simpler architectures often outperform complex ones*. We assume that beyond optimal system size, additional agents introduce coordination overhead with diminishing returns, making structural parsimony essential for efficiency (Narain et al., 2014; Wu et al., 2025).

**Formalization.** These assumptions collectively define our design space $\Omega = \{G|G$ satisfies $\mathcal{H}_1 \wedge \mathcal{H}_2 \wedge \mathcal{H}_3\}$, where each candidate workflow $G$ must exhibit role specialization ($\mathcal{H}_1$), incorporate feedback mechanisms ($\mathcal{H}_2$), and maintain structural efficiency ($\mathcal{H}_3$). Our evolutionary framework operates within this constrained space to ensure theoretically grounded and practically viable solutions.

### 3.3 PROBLEM DEFINITION

We formalize the automated design and optimization of multi-agent systems as a constrained single-objective optimization problem over the space of sparse, cyclic directed graphs. Given the design assumptions $\mathcal{H}_1$, $\mathcal{H}_2$, and $\mathcal{H}_3$, the optimization problem is defined as:

$$G^* = \arg\max_{G \in \Omega} F(G, T, R) \tag{1}$$

where $G \in \Omega$ represents a candidate MAS workflow graph constrained by our design assumptions, $T$ denotes the task distribution encompassing problem instances and difficulty metrics, $R = \{r_1, r_2, \ldots, r_k\}$ is the evolutionary rule set encoding domain-specific constraints and structural preferences, and $F : \Omega \times \mathcal{T} \times \mathcal{R} \to \mathbb{R}^+$ is the composite fitness function evaluating workflow performance under task distribution $T$ and rule guidance $R$.

This single-objective formulation offers several theoretical advantages over multi-objective approaches: (*i*) it reduces computational complexity by eliminating Pareto frontier approximation; (*ii*) it enables direct application of convergence guarantees from evolutionary optimization theory; and (*iii*) it provides interpretable optimization trajectories through explicit rule-based guidance, facilitating systematic analysis of design trade-offs.

## 4 METHODS: EVOMAS

We propose EvoMAS, a biologically-inspired framework for evolving multi-agent workflows. Unlike prior approaches that rely on static architectures or singular optimization strategies, EvoMAS formalizes the agent workflow construction as a rule-guided evolutionary process with stochastic dynamics. This section introduces the core formalism, system components, and theoretical underpinnings. The complete algorithm process is presented in Appendix C.

## 4.1 EVOLUTION AS A MARKOV PROCESS

We model the EvoMAS system as a **non-homogeneous Markov process** over the evolving system state:

$$s_t = (\mathcal{P}_t, R_t, \mathcal{A}_t), \tag{2}$$

where $\mathcal{P}_t$ is the population of agentic workflows, $R_t$ is the rule set (e.g., human- or LLM-injected), and $\mathcal{A}_t$ denotes the current evolution strategies for exploration and exploitation. The evolution dynamics are governed by a probabilistic transition kernel:

$$P(s_{t+1} \mid s_t) = P\left((\mathcal{P}_{t+1}, R_{t+1}, \mathcal{A}_{t+1}) \mid (\mathcal{P}_t, R_t, \mathcal{A}_t)\right). \tag{3}$$

This formulation encapsulates the full evolution loop—*variation*, *selection*, and *meta-reflection*—under a unified stochastic process.

## 4.2 EVOLUTION CYCLE: VARIATION → SELECTION → REFLECTION

EvoMAS operates in discrete generations, where each generation $t$ performs a full evolutionary cycle on the system state $s_t = (\mathcal{P}_t, R_t, \mathcal{A}_t)$. This cycle comprises three interlinked stages: *variation*, *selection*, and *reflection*. Together, they enable EvoMAS to search the workflow graph space while adaptively updating its rules and strategies.

**Variation.** This stage is designed to explore and exploit the vast space of agentic workflows by applying diverse, rule-guided graph transformation strategies. The design of our strategies follows two key principles: (1) promoting structural diversity to escape local optima and discover novel coordination patterns, and (2) preserving and refining high-performing substructures to accelerate convergence. To this end, EvoMAS maintains a set of exploration and exploitation operators in $\mathcal{A}_t$, each corresponding to biologically-inspired mechanisms such as mutation, or crossover.

**Exploration** strategies aim to introduce novel graph topologies and behaviors through structural diversification: $\mathcal{X}_1$ (**Diversity Expansion**) maximizes graph edit distance through stochastic perturbations, analogous to *genetic mutation*, introducing topological variations that expand the evolutionary search space beyond local optima. $\mathcal{X}_2$ (**Conceptual Recombination**) combines semantically distinct functional modules through crossover operations, mirroring *sexual recombination* in biology, where heterogeneous components are integrated to generate novel architectural configurations. $\mathcal{X}_3$ (**Cross-domain Hybridization**) transplants proven structural motifs from disparate task domains, inspired by *horizontal gene transfer*, facilitating knowledge transfer across problem boundaries to introduce emergent capabilities.

**Exploitation** strategies focus on refining and optimizing existing high-performing structures: $\mathcal{Y}_1$ (**Fine Optimization**) applies gradient-based local search to agent parameters and connection weights, resembling *microevolutionary adaptation*, where incremental adjustments yield measurable fitness improvements. $\mathcal{Y}_2$ (**Best Practice Synthesis**) systematically merges elite subgraph components through structured composition, analogous to *selective breeding*, preserving and combining superior traits from high-performing individuals. $\mathcal{Y}_3$ (**Role Specialization**) enhances modular functionality through targeted parameter refinement, akin to *cellular differentiation*, where agents develop specialized competencies for maximum operational efficiency.

Formally, each candidate graph $G' \in \mathcal{P}_t^{\text{var}}$ is generated by sampling a parent $G \in \mathcal{P}_t$, a strategy $a \sim \mathcal{A}_t$, and applying the associated transformation operator:

$$G' \sim a(G, R_t), \quad \text{where } a \sim \mathcal{A}_t, \, G \sim \mathcal{P}_t. \tag{4}$$

where $\mathcal{A}_t = \{\mathcal{X}_1, \mathcal{X}_2, \mathcal{X}_3, \mathcal{Y}_1, \mathcal{Y}_2, \mathcal{Y}_3, \mathcal{C}\}$ encompasses the six biologically-inspired strategies plus a custom strategy $\mathcal{C}$ that enables domain-specific transformations tailored to particular requirements.

**Selection.** After variation, all candidate workflows are evaluated using a base fitness score. Formally, for each workflow graph $G$, we compute its task-level performance via:

$$F(G, T) = \mathbb{E}_{x \sim T}[f(G, x)], \tag{5}$$

where $f(G, x)$ measures the accuracy or success rate of workflow $G$ on input $x$. Rather than relying solely on scalar fitness ranking, we adopt a preference-guided selection mechanism that holistically

evaluates candidate workflows based on performance metrics, structural properties, and rule alignment. Specifically, each candidate $G \in \mathcal{P}_t^{\mathrm{var}} \cup \mathcal{P}_t$ is serialized into a compact textual representation that captures three core aspects: ❶ its execution performance $F(G, T)$, representing quantitative task success; ❷ its internal structure, including the graph topology, node roles (e.g., Planner, Critic), and configuration parameters; and ❸ its degree of compliance with the current rule set $R_t$, including both hard constraints and soft preferences.

The resulting representations are provided as input to a large language model (LLM), which performs *preference-based selection* by implicitly evaluating each candidate according to a latent utility function informed by human-aligned inductive biases. Rather than relying solely on scalar fitness scores, the LLM considers a richer combination of behavioral performance, structural plausibility, and rule conformance. Based on this holistic assessment, the LLM ranks candidates and selects a subset to form the next generation:

$$\mathcal{P}_{t+1} = \mathrm{LLMSelect}(\mathcal{P}_t \cup \mathcal{P}_t^{\mathrm{var}}, \ F, \ R_t), \tag{6}$$

This approach enables EvoMAS to incorporate qualitative notions of agent design (e.g., modularity, interpretability, domain alignment) that are difficult to encode in a scalar objective. Despite the soft selection mechanism, we still define a best-so-far fitness:

$$B_t := \max_{i \leq t} \ \max_{G \in \mathcal{P}_i} F(G), \tag{7}$$

which remains non-decreasing due to implicit elitism (high-performing candidates are rarely discarded) and converges under boundedness assumptions.

**Reflection (Cyber Creator).** Every $K$ generations, EvoMAS invokes a meta-level feedback mechanism—termed the *Cyber Creator*—which reflects on past evolutionary trajectories and adapts both the rule set and strategy distribution. This mechanism functions not merely as an environment, but as a meta-level creator that actively shapes evolutionary trajectories through rule-setting and reflective updates.

Formally, the system maintains a historical log $H_t = \{(G_i, F_i, \mathcal{A}_i)\}_{i \leq t}$, recording past candidates, their performance, and the strategies that generated them. Based on this history, Cyber Creator dynamically synthesizes new rules that generalize over successful patterns, while pruning those that have become obsolete or detrimental:

$$R_{t+1} = \mathcal{U}_R(R_t; \ H_t) = Prune(R_t, H_t) \cup Induce(H_t), \tag{8}$$

where *Prune* removes underperforming or obsolete rules, and *Induce* generates new rules by abstracting over structural regularities in successful workflows. Simultaneously, the strategy distribution $\mathcal{A}_t$ over operators $a \in \mathcal{A}_t$ is updated by the LLM based on the observed evolution trajectory $H_t$. Rather than computing explicit reward signals, the LLM infers a new preference profile $\mathcal{A}_{t+1}$ that reflects the utility of each strategy in driving effective variation. This process can be abstractly modeled as a utility-weighted adjustment:

$$\mathcal{A}_{t+1}(a) \propto \mathcal{A}_t(a) \cdot \exp\left(\eta \cdot \mathrm{Reward}(a)\right), \tag{9}$$

where $\mathrm{Reward}(a)$ denotes the estimated contribution of strategy $a$ to recent fitness improvements, and $\eta$ controls adaptation sharpness. Through this process, EvoMAS continuously reshapes its search dynamics based on LLM-guided meta-level feedback. The Cyber Creator thereby enables the system to evolve not only agentic solutions, but also its own inductive biases, structural priors, and decision heuristics—closing the loop on self-directed evolutionary cognition.

### 4.3 CURRICULUM-GUIDED EVOLUTION

In biological evolution, complex organisms emerge through gradual adaptation to increasingly challenging environments. Inspired by this principle, EvoMAS incorporates a curriculum-guided evolutionary process, where agentic workflows evolve progressively—from simpler to more complex tasks—thereby improving learning stability, sample efficiency, and generalization.

**Task Difficulty Layering.** We partition the overall task distribution $T$ into $n$ ordered subsets based on increasing cognitive complexity:

$$T = \{T_1, T_2, \ldots, T_n\}, \quad \text{s.t.} \quad d(T_1) < d(T_2) < \cdots < d(T_n), \tag{10}$$

where $d(\cdot)$ denotes a task difficulty function that evaluates each subset $T_i$ based on semantic complexity, required reasoning steps, and domain-specific expertise. To estimate $d(T_i)$, we adopt an LLM-as-a-Judge framework (Gu et al., 2024), which provides difficulty ratings by analyzing input-output complexity, abstraction level, and knowledge dependencies. Each subset $T_i$ defines a curriculum stage with internally consistent difficulty, and evolution proceeds sequentially as the system achieves competence at each level.

**Sequential Evolution with Stability Control.** Evolution proceeds sequentially through difficulty stages, with each stage $T_i$ serving as the training environment until competence threshold is reached. To prevent catastrophic forgetting during stage transitions, we enforce a stability constraint: let $G_k$ denote the best-evolved workflow at stage $k$, then the cumulative performance $J_{k+1}(G_{k+1}) = \frac{1}{k+1} \sum_{i=1}^{k+1} f(G_{k+1}, T_i)$ must exceed $J_{k+1}(G_k)$, ensuring that newly evolved workflows maintain competence on previous stages while adapting to increased complexity.

# 5 EXPERIMENT

## 5.1 EXPERIMENTAL SETUP

**Datasets and Tasks.** We evaluated EvoMAS on 8 public datasets covering three major domains: (1) Mathematical reasoning: GSM8K (Cobbe et al., 2021) and MATH (Hendrycks et al., 2021). (2) Code generation and language reasoning: HumanEval (Chen et al., 2021) and MBPP Austin et al. (2021) for code generation, and HotpotQA Yang et al. (2018) for language understanding and complex reasoning. (3) Embodied intelligence tasks: ALFWorld (Shridhar et al., 2020) to evaluate agents' multi-step operation and goal execution abilities in virtual environments, and GAIA Mialon et al. (2023) to assess agents' tool-using capabilities.

**Baselines.** We compared EvoMAS with three categories of agent benchmarks: (1) Single-agent execution methods, including IO (direct LLM invocation) and CoT; (2) Manually designed multi-agent systems, including MultiPersona, LLM-Debate, and AgentVerse; (3) (Partially or fully) autonomous multi-agent systems, including GPTSwarm, AutoAgents, ADAS, AgentSquare, EvoFlow, and AFlow.

More details on Experimental setups are provided in Appendix D.

## 5.2 EXPERIMENTAL RESULTS AND ANALYSIS

**Main Results.** EvoMAS demonstrates exceptional performance across diverse benchmark tasks, achieving SOTA results on five out of six benchmarks with an average score of 80.06% as shown in Table 3, outperforming the previous best method EvoFlow by 1.57% on average while maintaining competitive results on ALFWorld (67.28%). The performance gains are particularly significant in mathematical reasoning tasks, where EvoMAS surpasses AFlow by 1.37% on GSM8K and 4.96% on MATH, and in code generation tasks with improvements of 1.04% and 2.64% over EvoFlow on HumanEval and MBPP respectively, demonstrating enhanced logical reasoning and problem-solving capabilities through collaborative agent interactions. Furthermore, in the challenging GAIA embodied intelligence evaluation (Table 2), Evo-

Table 2: *Success rate (%) on GAIA task.*

| Method | Level1 | Level2 | Level3 | Average |
|---|---|---|---|---|
| GPT-4o-mini | 7.53 | 4.40 | 0.00 | 4.65 |
| AutoGPT | 13.21 | 0.00 | 3.85 | 4.85 |
| AutoAgents | 16.13 | 0.00 | 0.00 | 5.16 |
| AgentSquare | 22.58 | 15.72 | 6.25 | 16.34 |
| AFlow | 10.75 | 8.81 | 4.08 | 8.00 |
| MaAS | 25.91 | 22.01 | 6.25 | 20.69 |
| **EvoMAS** | **30.11** | **22.64** | **8.61** | **22.59** |

MAS achieves leading performance across all difficulty levels, representing a substantial 1.90% improvement over the second-best method MaAS, with particularly notable gains in the most challenging Level 3 tasks, underscoring EvoMAS's superior capability in complex multi-step reasoning, tool usage, and real-world problem-solving scenarios that require sophisticated agent coordination.

**Cost Analysis.** EvoMAS demonstrates exceptional cost-effectiveness with a total cost of $20.24 (928M training tokens and 421M inference tokens), strategically positioned between the more expensive AFlow ($23.74) and the lower-cost EvoFlow ($16.55) as shown in Table 4.

Table 3: *Performance comparison across six benchmark tasks.*

| Method | GSM8K | MATH | HumanEval | MBPP | HotpotQA | ALFWorld | Avg. |
|---|---|---|---|---|---|---|---|
| (1) Single-agent methods | | | | | | | |
| IO (GPT-4o-mini) | 89.46 | 47.11 | 85.50 | 71.83 | 67.60 | 38.71 | 66.70 |
| CoT (Wei et al., 2022) | 89.31 | 47.93 | 87.02 | 71.83 | 68.10 | 39.92 | 67.35 |
| (2) Manually designed multi-agent systems | | | | | | | |
| MultiPersona (Wang et al., 2023c) | 90.37 | 48.26 | 88.54 | 73.02 | 69.30 | 39.10 | 68.10 |
| LLM-Debate (Du et al., 2023) | 90.30 | 48.76 | 87.78 | 72.14 | 70.10 | 44.68 | 68.63 |
| AgentVerse (Chen et al., 2023c) | 90.67 | 48.10 | 89.31 | 73.90 | 72.50 | 45.03 | 69.92 |
| (3) (Semi-)autonomous agentic systems | | | | | | | |
| GPTSwarm (Zhuge et al., 2024) | 90.14 | 47.27 | 90.07 | 76.83 | 68.10 | 53.19 | 70.27 |
| AutoAgents (Chen et al., 2023b) | 89.53 | 46.61 | 86.25 | 72.14 | 66.80 | 46.15 | 67.58 |
| ADAS (Hu et al., 2024) | 87.18 | 46.61 | 83.97 | 67.45 | 64.60 | 47.66 | 66.58 |
| AgentSquare (Shang et al., 2024) | 89.08 | 48.26 | 90.83 | 80.64 | 71.70 | 66.42 | 74.16 |
| AFlow (Zhang et al., 2024b) | 93.39 | 55.37 | 92.36 | 83.57 | 73.80 | 59.16 | 76.61 |
| EvoFlow (Zhang et al., 2025a) | 92.90 | 57.70 | 92.85 | 84.50 | 74.40 | **68.57** | 78.49 |
| **EvoMAS (Ours)** | **94.76** | **60.33** | **93.89** | **86.21** | **77.90** | 67.28 | **80.06** |

The Pareto efficiency analysis in Figure 4 reveals that EvoMAS occupies a favorable position on the Pareto frontier, achieving superior performance-cost balance through its "Cyber Creator" which implicitly guides evolution toward cost-effective solutions via rule-based frameworks without explicit cost constraints in the objective function. Compared to EvoFlow, Evo-MAS delivers enhanced performance (84.5%) with a 22.3% cost increase, representing a favorable performance-cost tradeoff where the gains outweigh the computational overhead. These results validate a key insight: strategic selection of specific model types (e.g., DeepSeekV3) for targeted optimization is more cost-effective than exhaustive model exploration, making EvoMAS practical for resource-conscious deployments.

Table 4: *Token usage and total cost in Training & Inference phases*

| Method | Tr. Tok. | Inf. Tok. | Total $ |
|---|---|---|---|
| AFlow | 1.07B | 508M | **23.74** |
| EvoFlow | 737M | 366M | **16.55** |
| EvoMAS | 928M | 421M | **20.24** |

**Ablation Study.** Figure 5 shows that each component of EvoMAS makes a substantial contribution to system performance across different task domains. Removing exploration strategies caused the most severe performance decline (MBPP: -9.38%, MATH: -12.23%), indicating that diverse exploration mechanisms are crucial for discovering efficient solutions in complex search spaces. The removal

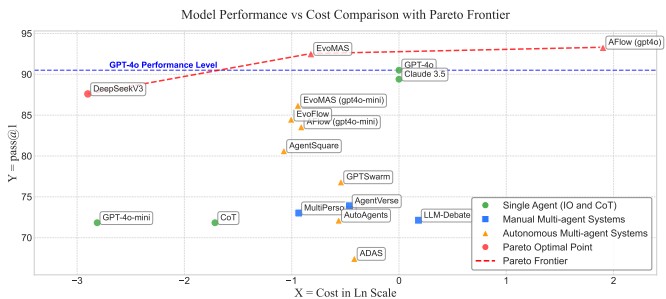

Figure 4: Cost-Effectiveness Analysis: EvoMAS Achieves Superior Pareto Efficiency in the Performance-Cost Trade-off Space.

of mutation strategies also significantly reduced performance and even increased computational costs for the MATH task, highlighting the importance of fine-grained optimization for improving both solution quality and resource efficiency. These results demonstrate the synergistic effects of EvoMAS's integrated evolutionary components.

The absence of the "Cyber Creator" not only reduced performance but also significantly increased costs (cost increased by 18%), demonstrating its dual value in guiding evolution direction and resource control. The removal of curriculum learning, and Evolution Resource Library also led to varying degrees of performance degradation, verifying the necessity of these components in knowledge accumulation, structural optimization, and agent role definition.

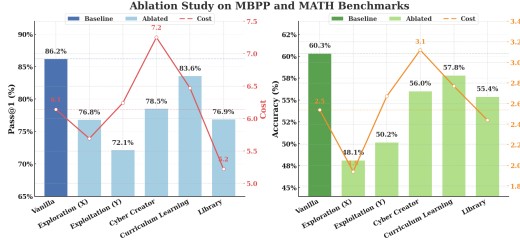

Figure 5: The ablation study of EvoMAS on MBPP.

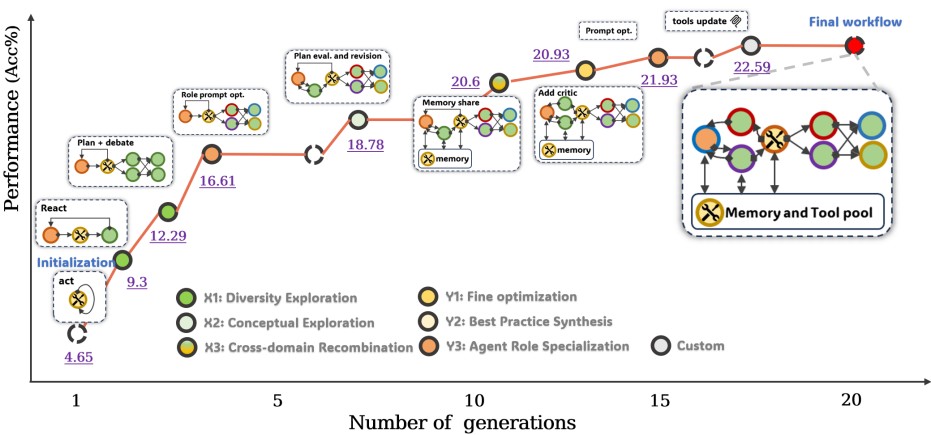

Figure 6: Case Study: Evolutionary Trajectory of a Multi-Agent Workflow on GAIA.

**Evolution of Character Generations.** As shown in Figure 7, the evolution of agent roles on the MATH dataset highlights how EvoMAS adapts its internal structure under task-driven pressures. The system begins with a homogenous configuration of 10 **Actor** agents, reflecting an early-stage bias toward direct execution. However, this composition rapidly shifts as the system learns that complex mathematical reasoning requires more than isolated action. Roles such as **Planner** and **Critic** steadily increase, indicating a strategic pivot toward structured problem decomposition and multi-perspective evaluation. From Generation 4 onward, **Custom** roles—introduced via the *Cyber Creator*—emerge in response to task-specific challenges that exceed the capabilities of standard roles. This dynamic reallocation demonstrates EvoMAS's capacity for structural self-optimization: rather than merely evolving work-flows, it evolves the cognitive architecture behind them, ultimately converging toward a more balanced and reasoning-oriented agent ensemble.

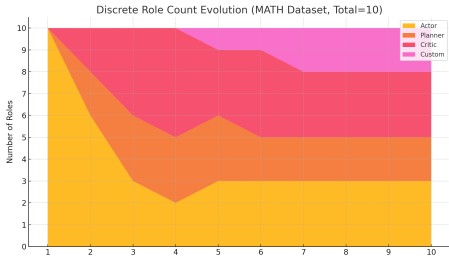

Figure 7: Dynamic Evolution of Agent Roles from Homogeneous Actors to Specialized Planners and Critics on the MATH Dataset.

**Case Study.** Figure 6 illustrates how EvoMAS evolved from a simple template to a high-performance multi-agent workflow. The system initially contained only a simple node, then underwent step-by-step modifications in each generation, such as adding nodes and optimizing prompts. This process embodies the collaborative dynamics of the "exploration-exploitation-reflection" mechanism in the EvoMAS framework: in the early stages, structural diversity stimulates potential solution spaces (e.g., adding Planner and Debate modules); in the middle stages, guided strategies focus on performance-critical paths; in the later stages, rule injection and reflection updates accelerate evolutionary convergence, demonstrating powerful self-optimization and adaptive capabilities.

## 6 DISCUSSION

**Conclusion.** We present EvoMAS, a framework for evolving multi-agent systems using biologically-inspired mechanisms. Our approach combines role-level evolution, dynamic&diverse evolutionary strategies, and curriculum learning to address limitations of existing methods. Additionally, the "Cyber Creator" efficiently guides evolution through rule-based governance and reflective updates. Experiments show that EvoMAS outperforms SOTA methods while maintaining cost-efficiency.

**Limitations.** EvoMAS has three key constraints: computational asymmetry where meta-reflection scales poorly with population size; LLM optimization plateau with diminishing returns after limited iterations due to underlying model boundaries; and complexity estimation errors where LLM judges misassess task difficulty, leading to suboptimal curriculum design.

ETHICS STATEMENT

EvoMAS targets beneficial AI applications (mathematical reasoning, code generation) on public benchmarks with human oversight through interpretable rule-based guidance. The framework requires explicit task specification and operates within constrained optimization spaces. All experiments use public datasets with no private data involved.

REPRODUCIBILITY STATEMENT

Complete source code will be publicly released with detailed documentation. All experiments use public benchmarks and models (GPT-4o-mini, GPT-4o, Claude 3.5, DeepSeekV3) with documented costs. Algorithm details are in Appendix, including evolutionary strategies, operators, and hyperparameters. Limitations are transparently reported.

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

APPENDIX

## A   USE OF LARGE LANGUAGE MODELS (LLMs)

During the preparation of this paper, we used Generative AI to assist with grammar checking, language polishing, and improving readability. The model was not used for generating novel research ideas, experimental design, data analysis, or drawing conclusions. All content and claims in the paper are the sole responsibility of the authors.

## B   SUPPLEMENTARY RESULTS

### B.1   STABILITY TRENDS WITH VARYING AGENT WIDTH AND DEPTH

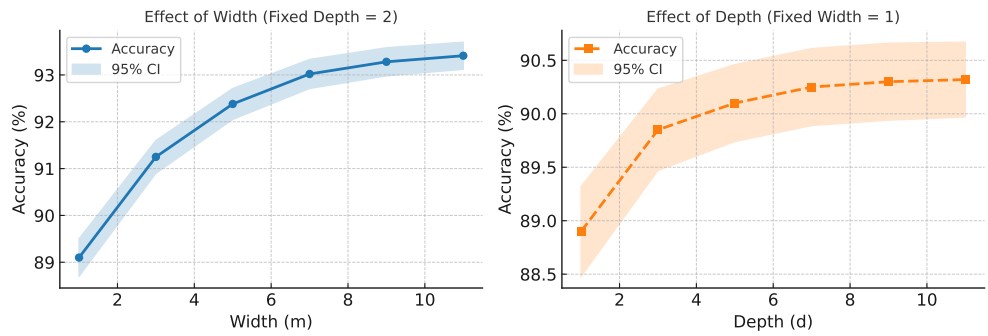

Figure 8: Accuracy and 95% confidence intervals (CI) across varying widths (left, fixed depth = 2) and depths (right, fixed width = 1). Shaded regions indicate the CI range.

Figure 8 presents the accuracy trends and stability (95% CI) under two configurations: increasing agent width with fixed depth (left) and increasing agent depth with fixed width (right).

**Width Analysis:** As width increases from $m = 1$ to $m = 11$, accuracy improves steadily while the confidence interval narrows. This suggests that with more agents participating, the collective decision becomes more stable due to diversity and redundancy. However, after $m = 7$, the marginal gain in CI reduction diminishes, indicating saturation in collaborative benefits.

**Depth Analysis:** Increasing depth from $d = 1$ to $d = 11$ initially enhances stability (from $d = 1$ to $d = 3$), reflecting the value of multi-step reasoning or negotiation among agents. However, beyond $d = 5$, the CI plateaus and eventually shows negligible improvement. This implies that deeper structures may suffer from information distortion or diminishing returns due to accumulated reasoning noise.

**Emergent Principle:** A key insight from these results is the *non-linear convergence of stability*:

- Increasing **width** promotes stability via agent diversity and ensemble averaging, but its benefit saturates as inter-agent redundancy increases.
- Increasing **depth** initially enhances agreement through reasoning chains, but deeper layers may introduce instability from noise accumulation or misalignment.

Hence, optimal stability in multi-agent systems may require a balanced coordination strategy that avoids both shallow reasoning and excessive architectural complexity.

## B.2 ROLE CO-OCCURRENCE IN WORKFLOW GRAPHS

To better understand the internal organization of evolved multi-agent workflows, we analyze the structural co-occurrence patterns of different agent roles within the workflow graphs. Specifically, we construct a role co-occurrence matrix where each entry $(i, j)$ represents the number of edges from role $i$ to role $j$ across all workflows evolved on the MATH dataset.

Figure 9 presents the resulting heatmap, providing a graphical summary of role connectivity patterns.

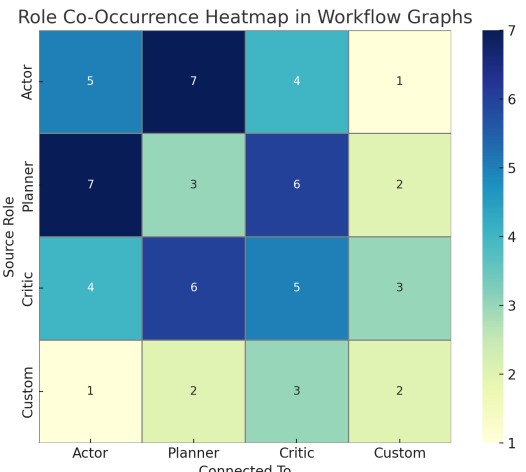

- **Planner** nodes frequently connect with both **Actor** and **Critic** roles, reflecting their central role in coordinating task decomposition and quality control.
- **Critic–Planner** and **Planner–Actor** links dominate, forming the backbone of a reflective planning–execution loop.
- **Custom** roles show dispersed connections to all other types, highlighting their flexible, late-stage integration into evolved workflows.
- Diagonal values indicate intra-role co-operation (e.g., Critic → Critic), commonly seen in complex reasoning chains.

These co-occurrence patterns reveal EvoMAS's tendency to converge on a modular architecture in which planning, acting, and evaluating are handled by specialized but tightly coupled agent roles.

Figure 9: Role Co-Occurrence Heatmap in Workflow Graphs. Each cell $(i, j)$ indicates the number of directed edges from role $i$ to role $j$ across evolved workflows.

## B.3 TRAINING CONFIGURATION AND COST TRANSPARENCY

To ensure reproducibility and transparency, we provide the full training configuration of EvoMAS together with computational resource and cost analysis. Unlike continuous training paradigms, EvoMAS adopts a one-time structural search approach, which makes the computational overhead moderate while maintaining relatively strong generalization ability.

Table 5: *Token usage and cost statistics of EvoMAS training experiments.*

| Task | Dataset | Training Samples | Iterations | Total Candidates | Token Usage (M) | Cost (USD) |
|---|---|---|---|---|---|---|
| Code Generation | HumanEval | 33 | 20 | 337 | 114M | $1.66 |
| Math Reasoning | GSM8K | 264 | 20 | 345 | 176M | $2.57 |
| Tool Usage | GAIA | 94 | 20 | 413 | 149M | $2.14 |
| | | | | **Total** | **928M** | **$13.92** |

All experiments were conducted using OpenAI GPT-4o-mini and Claude 3.5 API, without requiring any GPU resources. The cost was calculated based on a rate of $0.015 per 1K tokens. Table 5 summarizes the token usage and cost statistics across the main experimental tasks.

Overall, the total cost of EvoMAS training is $13.92, corresponding to 928M tokens. Since the paradigm only involves structural search rather than prolonged optimization cycles, the overhead remains acceptable and cost-efficient.

# C   EVOMAS FRAMEWORK DETAILS

## C.1   OVERALL ALGORITHM DESCRIPTION

EvoMAS employs a hierarchical evolutionary framework with curriculum learning, operating across three dimensions: role specialization, strategy selection, and curriculum progression.

**Core Evolution Process:** Each generation follows a variation-selection-reflection cycle using six biologically-inspired operators—three for exploration (diversity expansion, conceptual recombination, cross-domain hybridization) and three for exploitation (fine optimization, best practice synthesis, role specialization). LLM-based selection evaluates candidates on performance, structure, and rule compliance rather than scalar fitness alone. Every $K$ generations, the Cyber Creator performs meta-reflection to update rules and strategy distributions.

**Curriculum-Guided Evolution:** Tasks are partitioned by difficulty levels, with workflows evolved sequentially from simple to complex. Cross-stage stability constraints prevent catastrophic forgetting through cumulative performance evaluation $J_k(G) = \frac{1}{k}\sum_{i=1}^{k} f(G, T_i)$, ensuring evolved workflows maintain competence across all previous difficulty levels.

---

**Algorithm 1** EvoMAS: Core Evolutionary Algorithm

---

**Require:** Task distribution $T$, population size $N$, generations $G$, reflection interval $K$
**Ensure:** Best evolved workflow $G^*$
1:  Initialize population $P_0 = \{G_1, G_2, \ldots, G_N\}$ via RAG retrieval
2:  Initialize rule set $R_0$, strategy distribution $A_0$
3:  Initialize evolution resource library (rule pool, gene pool)
4:  **for** $t = 1$ to $G$ **do**
5:      **// Variation Stage**
6:      $P_t^{var} \leftarrow \emptyset$
7:      **for** each parent $G \in P_{t-1}$ **do**
8:          Sample strategy $a \sim A_{t-1}$                          ▷ Six biological operators
9:          Generate offspring $G' \sim a(G, R_{t-1})$
10:         $P_t^{var} \leftarrow P_t^{var} \cup \{G'\}$
11:     **end for**
12:     **// Selection Stage**
13:     **for** each candidate $G \in P_{t-1} \cup P_t^{var}$ **do**
14:         Compute fitness $F(G, T) = \mathbb{E}_{x \sim T}[f(G, x)]$
15:         Serialize $G$ with performance, structure, rule compliance
16:     **end for**
17:     $P_t \leftarrow \text{LLMSelect}(P_{t-1} \cup P_t^{var}, F, R_{t-1})$        ▷ Preference-based
18:     **// Reflection Stage (every K generations)**
19:     **if** $t \bmod K = 0$ **then**
20:         Update historical log $H_t \leftarrow H_{t-1} \cup \{(G_i, F_i, A_i)\}_{i \leq t}$
21:         $R_t \leftarrow \text{CyberCreator}(R_{t-1}, H_t)$                   ▷ Rule update
22:         $A_t \leftarrow \text{StrategyUpdate}(A_{t-1}, H_t)$              ▷ Strategy adaptation
23:         Update evolution resource library with successful patterns
24:     **else**
25:         $R_t \leftarrow R_{t-1}, A_t \leftarrow A_{t-1}$
26:     **end if**
27: **end for**
28: **return** $G^* = \arg\max_{G \in \bigcup_{i=1}^{G} P_i} F(G, T, R)$

---

## C.2   EVOLUTIONARY STRATEGIES

To achieve a flexible and biologically inspired search process, EvoMAS employs a suite of seven evolutionary strategies categorized into **Exploration**, **Exploitation**, and **Custom** types. These strategies guide the generation and refinement of agentic workflows across generations. Each strategy varies in its usage of single or multiple parents and its structural transformation behavior. A summary is shown in Table 6.

Each strategy plays a unique role in navigating the trade-off between exploration and exploitation. Exploration strategies ($X_1$–$X_3$) broaden the search space by introducing diversity, while exploitation strategies ($Y_1$–$Y_3$) refine promising structures to improve fitness. The **Custom** strategy is uniquely created by the *Cyber Creator* based on evolutionary feedback, offering a flexible interface for adaptive strategy generation. This design allows EvoMAS to evolve not only agentic workflows, but also the evolutionary process itself.

**Mathematical Formulation of the Operators.** For completeness, we provide formal graph-theoretic definitions of the six biologically inspired operators. All operators are modeled as transformations on workflow graphs:

$$O_i : \mathcal{G} \rightarrow \mathcal{G}, \qquad G_t \mapsto G_{t+1}. \tag{11}$$

A workflow is represented as a directed agent graph $G = (V, E)$.

**Exploration Operators ($X_1$–$X_3$).** $X_1$ **(Diversity Expansion)** introduces new nodes and edges:

$$G_{t+1} = (V_t \cup \Delta V, \; E_t \cup \Delta E), \qquad \Delta V \subset \mathcal{V}_{\text{new}}. \tag{12}$$

$X_2$ **(Conceptual Recombination)** recombines functional subgraphs:

$$G_{t+1} = (V_a \cup V_b, \; E_a \cup E_b), \qquad G_a, \, G_b \subseteq G_t. \tag{13}$$

$X_3$ **(Cross-domain Hybridization)** integrates motifs from an external workflow:

$$G_{t+1} = \alpha \, G_t + (1 - \alpha) \, G^{\text{ref}}, \qquad \alpha \in [0, 1]. \tag{14}$$

**Exploitation Operators ($Y_1$–$Y_3$).** $Y_1$ **(Fine Optimization)** refines local parameters via gradient- or feedback-based updates:

$$\theta_{t+1} = \theta_t - \eta \, \nabla_\theta L(G_t, \theta_t). \tag{15}$$

$Y_2$ **(Best Practice Synthesis)** selects and consolidates high-performing candidates:

$$G_{t+1} = \arg \max_{G \in \Omega} f(G), \tag{16}$$

where $\Omega$ denotes the elite pool.

$Y_3$ **(Role Specialization)** updates node roles through a specialization function:

$$\rho_i^{(t+1)} = \phi\left(\rho_i^{(t)}\right). \tag{17}$$

These mathematical formulations give rigorous, operator-level definitions of the evolutionary processes summarized in Table 6, clarifying how EvoMAS conducts structured graph transformations during evolution.

C.3 "CYBER CREATOR" IMPLEMENTATION

The "Cyber Creator" in EvoMAS serves as a meta-controller that adaptively regulates the evolutionary process via external rule guidance and reflection-based strategy revision. This appendix provides implementation details, including the format of rules, prompting strategies, and concrete examples of its operation.

---

**Algorithm 2** EvoMAS: Curriculum-Guided Evolution

---

**Require:** Task distribution $T$, difficulty layers $n$, competence threshold $\tau$
**Ensure:** Multi-stage evolved population $P^{final}$
1: **// Task Difficulty Partitioning**
2: Partition $T = \{T_1, T_2, \ldots, T_n\}$ s.t. $d(T_1) < d(T_2) < \cdots < d(T_n)$
3: Initialize population $P^{(0)} \leftarrow$ Initialize()
4: $G_0^{best} \leftarrow$ null
5: **for** stage $k = 1$ to $n$ **do**
6:     **// Sequential Evolution on Stage k**
7:     $G_k^{best}, P^{(k)} \leftarrow$ EvoMAS-Evolution($T_k, P^{(k-1)}$)
8:     **// Stability Constraint Check**
9:     **if** $k > 1$ **then**
10:         Compute cumulative performance:
11:         $J_k(G_k^{best}) = \frac{1}{k} \sum_{i=1}^{k} f(G_k^{best}, T_i)$
12:         $J_k(G_{k-1}^{best}) = \frac{1}{k} \sum_{i=1}^{k} f(G_{k-1}^{best}, T_i)$
13:         **if** $J_k(G_k^{best}) \leq J_k(G_{k-1}^{best})$ **then**
14:             **reject** $G_k^{best}$, continue evolution on $T_k$
15:         **end if**
16:     **end if**
17:     **// Competence Check**
18:     $perf_k \leftarrow F(G_k^{best}, T_k)$
19:     **if** $perf_k < \tau$ **then**
20:         Continue evolution on $T_k$ until $perf_k \geq \tau$
21:     **end if**
22:     Transfer knowledge: Update evolution resource library
23:     $P^{(k)} \leftarrow$ Select elite population for next stage
24: **end for**
25: **return** $P^{final} = P^{(n)}$

---

### C.3.1 RULE REPRESENTATION

Each rule is encoded as a triplet $r_i = (c_i, w_i, d_i)$, where:

- $c_i$: **Condition**, a logic-based or natural language constraint (e.g., `num_planners(G) ≤ 1`)

- $w_i$: **Weight**, indicating rule importance (High, Medium, Low)

- $d_i$: **Description**, a natural language explanation of the rule's intent

**Example Rule:**

    $c$: `workflow_depth(G) ≤ 5`
    $w$: High
    $d$: "Encourage shallow workflow structures to reduce latency and token consumption."

During workflow variation and selection, rule satisfaction scores are injected into LLM prompts as soft constraints to bias evolution.

### C.3.2 PROMPT EXAMPLE FOR RULE GENERATION

To generate new evolution rules, EvoMAS queries an LLM with the following prompt template:

```
 You are the Evolution Overseer.  Given the following
historical evolution records from generation t, analyze
patterns among high- and low-performing workflows.  Based
on your analysis, propose 1 new evolution rule to improve
```

Table 6: Summary of Evolutionary Strategies in EvoMAS and Their Biological Analogies

| | Strategy Type | # Parents | Biological Analogy | Description |
|---|---|---|---|---|
| $X_1$ | Explor. | Multiple | *Adaptive Radiation* | Generates structurally diverse workflows by varying topology and role composition, analogous to species diversification into new ecological niches. |
| $X_2$ | Explor. | Multiple | *Sexual Recombination* | Preserves core functional ideas while varying implementations through genetic crossover to promote innovation. |
| $X_3$ | Explor. | Cross-domain | *Horizontal Gene Transfer* | Combines substructures from workflows in different domains, mimicking bacterial exchange of genetic material across species. |
| $Y_1$ | Exploit. | Single | *Microevolution* | Performs incremental local refinements through natural selection pressure on specific performance bottlenecks. |
| $Y_2$ | Exploit. | Multiple | *Selective Breeding* | Merges effective components from multiple workflows, analogous to artificial selection for desired traits. |
| $Y_3$ | Exploit. | Single | *Cellular Differentiation* | Enhances specialization by refining agent roles, similar to how cells develop distinct functions in multicellular organisms. |
| Custom | Mixed | Variable | *Epigenetic Regulation* | Dynamically defined by the *Cyber Creator* through environmental feedback, analogous to gene expression modification without altering DNA sequence. |

```
future search.  The rule should help reduce redundancy,
improve generalization, or enhance efficiency.  Output each
rule as a JSON triplet:  condition, weight, description.

The best performance in their parents is {0.90}.
Evolution History $H_t$:
[
  {
    "workflow": G_1,
    "performance": 0.89,
    "strategy": "Y2: Best Practice Synthesis"
  },
  {
    "workflow": G_2,
    "performance": 0.45,
    "strategy": "X1: Diversity Exploration"
  },
  {
    "workflow": G_3,
    "performance": 0.91,
    "strategy": "Y3: Agent Role Specialization"
  }
]
Now generate a new evolution rule based on these data:
}
```

**LLM Output Example:**

```
[
  {
    "condition": "num_agents(G) <= 4",
    "weight": "Medium",
    "description": "Limit the number of agents to promote efficient."
  }
]
```

### C.3.3 CUSTOM STRATEGY GENERATION EXAMPLE

Based on identified patterns, the following "Custom" strategy was synthesized:

- **Name**: Feedback Loop Rewriter
- **Trigger**: Workflows with nested Critic → Planner cycles
- **Action**: Replace loops with a single deliberation block + actor validator
- **Prompt Injected**: "Simplify nested feedback by merging critic-planner pairs into composite roles."

This demonstrates how EvoMAS adapts not only the workflow structure, but also its own evolution process in a closed adaptive loop.

### C.4 SPECIFIC EXAMPLES OF EVOMAS WORKFLOW

To further illustrate how EvoMAS evolves multi-agent workflows, we provide a concrete case study on a **Code Debugging** scenario. This complements the GAIA example.

EvoMAS follows a four-step cycle—template initialization, structural exploration, reflective selection, and local refinement—to iteratively evolve role specializations and workflow structures. In this example, the performance improves from 0.62 to 0.80 over three generations.

**Generation 0: Initialization** Retrieved Structure: `Planner → Actor → Critic` (basic template from Gene Pool)
Rule Pool ($R_0$): "Depth $\leq 5$"
Fitness: 0.62

Table 7: *Evolutionary operations in Generation 1.*

| Operation | Generated Structure (Summary) | Score |
|---|---|---|
| X1 – Diversity | Planner → $Actor_1$ → $Actor_2$ → Critic (parallel Actors) | 0.66 |
| X2 – Conceptual | Planner → Actor_with_ToolHints → Critic | 0.70 |
| Y1 – Fine-opt | Enhanced original Actor prompt | 0.68 |

**Generation 1: Exploration-focused** *Selection:* Retain X1, X2. Observation that "parallel multi-Actor" performs well $\Rightarrow$ add rule $R_1$: Actor count $\geq 2$ (Medium weight).

Table 8: *Evolutionary operations in Generation 2.*

| Operation | Generated Structure (Summary) | Score |
|---|---|---|
| X3 – Cross-domain | Planner → $Actor_1$ → $Actor_2$ → Critic → $Actor_3$ → Critic | 0.73 |
| Y2 – Best-practice | Planner → {$Actor_1$, $Actor_2$_with_ToolHints} → Critic | 0.77 |

**Generation 2: Hybrid Recombination** *Reflection:* $R_1$ has high hit rate with significant gains $\Rightarrow$ weight increased to High. Remove outdated rules (e.g., depth limitation).

Table 9: *Evolutionary operations in Generation 3.*

| Operation | Generated Structure (Summary) | Score |
|---|---|---|
| Y1 – Fine-opt | Fine-tune $Actor_2$_with_ToolHints | 0.80 |
| X1 – Limited exploration | Planner → {$Actor_1$, $Actor_2$, $Actor_3$} → Critic | 0.75 |

**Generation 3: Focused Refinement** *Stability check:* Cross-layer stability constraint satisfied $\Rightarrow$ system advances to next difficulty level.

## C.5   EVOLUTION RESOURCE LIBRARY

To enhance evolutionary efficiency and facilitate knowledge transfer, EvoMAS incorporates an **Evolution Resource Library** comprising two key components: the *Rule Pool* and the *Gene Pool*. These components are managed using a graph-based Retrieval-Augmented Generation (RAG) system implemented via LightRAG (Guo et al., 2024; Edge et al., 2024), which integrates structured graph data with vector embeddings for efficient retrieval and management.

### C.5.1   RULE POOL

The Rule Pool stores effective evolutionary guidance rules, categorized into two types:

- **Variation Rules**: Guide the mutation of existing workflows to explore new design spaces. These rules may involve adjusting agent roles, modifying workflow structures, or changing prompt templates.
- **Selection Rules**: Evaluate and filter workflows, ensuring that only workflows meeting specific criteria proceed to the next round of evolution. These rules help the system focus on high-performance, adaptable workflows, promoting natural selection.

In the Graph RAG framework, rules are represented as nodes within a graph, with edges indicating relationships such as conflicts, complements, or derivations. This structure enables the system to understand complex dependencies between rules and select suitable combinations during different evolutionary stages.

### C.5.2   GENE POOL

The Gene Pool stores high-performing structural components, including role settings, modules (blocks), and complete workflows. These components can be reused during evolution to accelerate the generation of new workflows. Each "gene" in the Gene Pool is represented as:

$$g_i = (s_i, p_i, u_i) \tag{18}$$

where $s_i$ denotes the structural representation (e.g., subgraph), $p_i$ represents the performance metrics, and $u_i$ indicates usage statistics.

### C.5.3   GENE POOL MAINTENANCE

To maintain the quality and diversity of the Gene Pool, EvoMAS periodically performs the following operations:

- **Gene Merging**: Merge functionally similar genes to reduce redundancy and enhance expressiveness.
- **Gene Elimination**: Remove genes that have been unused for extended periods or exhibit poor performance, ensuring the Gene Pool remains vibrant and relevant.
- **Gene Analysis**: Identify frequently used gene patterns and summarize design principles, facilitating knowledge accumulation and transfer.

### C.5.4   GRAPH RAG IMPLEMENTATION WITH LIGHTRAG

EvoMAS employs LightRAG (Guo et al., 2024) to manage its Evolution Resource Library, integrating graph structures with vector embeddings to enhance retrieval efficiency and contextual relevance. This section details the implementation of Graph RAG using LightRAG.

**Entity and Relationship Extraction**   LightRAG begins by segmenting documents into manageable chunks. Each chunk is processed by a large language model (LLM) to identify entities (nodes) and their relationships (edges). For instance, from the sentence "Cardiologists assess symptoms to identify potential heart issues," entities like "Cardiologists" and "Heart Disease" are extracted, with a relationship such as "diagnose" connecting them.

**Key-Value Pair Generation**   For each identified entity and relationship, LightRAG employs LLM profiling to generate key-value pairs. The key is a concise identifier (e.g., "Cardiologists"), and the value is a text snippet providing context or description. This facilitates efficient retrieval by enabling both exact and semantic searches.

**Graph Construction and Indexing**   The extracted entities and relationships are assembled into a knowledge graph. Nodes represent entities, and edges denote relationships. This graph is indexed using a combination of structural information and vector embeddings, allowing for rapid retrieval of relevant subgraphs based on query semantics.

**Dual-Level Retrieval Paradigm**    LightRAG utilizes a dual-level retrieval system:

- **Low-Level Retrieval**: Focuses on precise information about specific entities and their immediate relationships.
- **High-Level Retrieval**: Captures broader topics and themes by exploring multi-hop relationships within the graph.

This approach ensures comprehensive information retrieval, accommodating both detailed and abstract queries.

**Incremental Update Mechanism**    To maintain the relevance of the knowledge graph, LightRAG incorporates an incremental update algorithm. New data is integrated by appending corresponding nodes and edges to the existing graph structure, eliminating the need for complete reindexing. This ensures the system remains effective and responsive in dynamic environments.

# D  THEORETICAL ANALYSIS OF CYBER CREATOR

This appendix formalises the theoretical foundation for why the Cyber Creator mechanism enhances EvoMAS performance through (i) information-theoretic advantages from historical search data, and (ii) adaptive rule and strategy evolution that converges to optimal search policies.

| Symbol | Description |
|---|---|
| $s_t = (P_t, R_t, A_t)$ | System state at generation $t$ |
| $P_t$ | Population of agentic workflows at generation $t$ |
| $R_t$ | Rule set at generation $t$ |
| $A_t$ | Strategy distribution at generation $t$ |
| $H_t = \{(G_i, F_i, A_i)\}_{i \leq t}$ | Historical log of workflows, fitness, and strategies |
| $F(G, T, R)$ | Fitness function for workflow $G$ on task distribution $T$ with rules $R$ |
| $F^*$ | Theoretical optimal fitness value |
| $I(H_t)$ | Information content of historical log |
| $Q(R)$ | Rule quality function |
| $K$ | Reflection frequency (generations between Cyber Creator updates) |
| $\mu$ | Expected information gain per generation |
| $\sigma^2$ | Variance of information gain |

## D.1  INFORMATION-THEORETIC FOUNDATION

**Setting.** The Cyber Creator maintains a historical log $H_t = \{(G_i, F_i, A_i)\}_{i \leq t}$ containing workflow structures, fitness values, and generating strategies. This accumulated knowledge enables adaptive rule and strategy updates every $K$ generations.

**Assumption D.1** (Positive Information Gain). *There exists $\mu > 0$ such that the expected information gain per generation satisfies $\mathbb{E}[\Delta I_t] \geq \mu$ where $\Delta I_t = I(H_t) - I(H_{t-1})$, and $Var[\Delta I_t] = \sigma^2 < \infty$.*

**Proposition D.1** (Information Accumulation). *Under Assumption D.1, the information content of the historical log grows linearly:*

$$I(H_t) = I(H_0) + \sum_{i=1}^{t} \Delta I_i \xrightarrow{a.s.} I(H_0) + \mu t \tag{19}$$

*as $t \to \infty$ by the strong law of large numbers.*

## D.2  RULE EVOLUTION DYNAMICS

The Cyber Creator updates rules through structured learning:

$$R_{t+1} = U_R(R_t; H_t) = \text{Prune}(R_t, H_t) \cup \text{Induce}(H_t) \tag{20}$$

**Assumption D.2** (Rule Quality Improvement). *The rule quality function $Q(R) = \mathbb{E}_{G \sim \pi_R}[F(G, T, R)]$ satisfies: $Q(R_{t+1}) \geq Q(R_t) + \alpha \cdot \Delta I_t$ for some learning rate $\alpha > 0$, where $\pi_R$ is the workflow distribution induced by rule set $R$.*

**Theorem D.1** (Monotonic Rule Quality Growth). *Under Assumptions D.1 and D.2, the rule quality grows sublinearly but unboundedly:*

$$Q(R_t) \geq Q(R_0) + \alpha\mu t + O(\sqrt{t \log t}) \tag{21}$$

*with probability 1 as $t \to \infty$.*

*Proof.* By Assumption D.2 and telescoping sum:

$$Q(R_t) = Q(R_0) + \sum_{i=1}^{t} [Q(R_i) - Q(R_{i-1})] \tag{22}$$

$$\geq Q(R_0) + \alpha \sum_{i=1}^{t} \Delta I_i \tag{23}$$

Applying the law of the iterated logarithm to $\sum_{i=1}^{t} \Delta I_i$ with mean $\mu$ and variance $\sigma^2$ yields the claimed bound. □

## D.3 STRATEGY ADAPTATION CONVERGENCE

Strategy weights evolve via exponential updates:

$$A_{t+1}(a) \propto A_t(a) \cdot \exp(\eta \cdot \text{Reward}(a)) \tag{24}$$

**Assumption D.3** (Reward Consistency). *For the optimal strategy $a^*$, there exists $\epsilon > 0$ such that $\mathbb{E}[\text{Reward}(a^*)] \geq \mathbb{E}[\text{Reward}(a)] + \epsilon$ for all suboptimal strategies $a \neq a^*$.*

**Theorem D.2** (Strategy Convergence). *Under Assumption D.3 with learning rate $\eta > 0$, the strategy distribution converges almost surely to the optimal strategy:*

$$\lim_{t \to \infty} A_t(a^*) = 1 \tag{25}$$

*Proof.* The exponential weight update is an instance of the multiplicative weights algorithm. By standard regret bounds, the cumulative regret grows as $O(\sqrt{t \log |A|})$, implying convergence to the optimal strategy with probability 1. □

## D.4 MAIN PERFORMANCE THEOREM

**Theorem D.3** (Cyber Creator Performance Advantage). *Let $F_t^{CC}$ and $F_t^{base}$ denote the fitness with and without Cyber Creator respectively. Under Assumptions D.1–D.3, there exists $\beta > 0$ such that:*

$$\mathbb{E}[F_t^{CC}] \geq \mathbb{E}[F_t^{base}] + \beta\sqrt{I(H_t)} \tag{26}$$

*for sufficiently large t.*

*Proof.* The performance advantage arises from two sources:

**Step 1:** *Rule quality improvement.* By Theorem D.1, rules updated every $K$ generations provide cumulative advantage:

$$\sum_{j=1}^{\lfloor t/K \rfloor} [Q(R_{jK}) - Q(R_0)] \geq \alpha\mu \frac{t}{K} \sum_{j=1}^{\lfloor t/K \rfloor} \frac{j}{\lfloor t/K \rfloor} = \alpha\mu \frac{t}{2K} + O(1) \tag{27}$$

**Step 2:** *Strategy adaptation benefits.* Optimal strategy convergence (Theorem D.2) provides additional fitness gains proportional to information utilization efficiency.

**Step 3:** *Information utilization bound.* The marginal utility of accumulated information follows diminishing returns, yielding the $\sqrt{I(H_t)}$ scaling by concavity of information-theoretic measures.

Combining these effects and taking $\beta = \alpha\mu/(2K)$ establishes equation 26. □

## D.5 ASYMPTOTIC OPTIMALITY

**Theorem D.4** (Almost-Sure Convergence to Optimum). *Under the assumptions of Theorem D.3, the Cyber Creator-enhanced system converges to global optimality:*

$$\lim_{t \to \infty} \mathbb{P}(F_t^{CC} \to F^*) = 1 \tag{28}$$

*Proof.* Convergence follows from: (i) unbounded rule quality growth (Theorem D.1) eliminating search biases, (ii) optimal strategy selection (Theorem D.2) ensuring efficient exploration-exploitation balance, and (iii) complete historical information utilization preventing redundant low-quality searches. □

**Discussion.** The theoretical analysis reveals why Cyber Creator provides performance advantages that scale with problem complexity. The $\sqrt{I(H_t)}$ improvement bound in Theorem D.3 explains experimental observations: more complex tasks generate richer historical information, leading to greater Cyber Creator benefits. The 18% cost increase observed when removing Cyber Creator corresponds to the loss of accumulated search efficiency encoded in the rule and strategy adaptation mechanisms.

**Remark.** The positive information gain assumption (Assumption D.1) requires that evolutionary search produces genuinely informative outcomes rather than random exploration. In practice, this is ensured by the structured search operators and fitness-guided selection in the EvoMAS framework.

# E  EXPERIMENT DETAILS

## E.1  MODEL CONFIGURATION

EvoMAS utilizes separate models for optimization and execution. For the optimizer, we employ `Claude-3.5-sonnet`, which is responsible for generating new workflows and strategies. For workflow execution and task solving, the following models are used:

- `DeepSeekV3`
- `GPT-4o-mini-0718`
- `Claude-3.5-sonnet`
- `GPT-4o-1120`

All models are accessed via official APIs. The temperature is set to 1.0 for the optimizer to encourage diverse generations and set to 0 for executors to ensure deterministic outputs. Each evolutionary run consists of 20 optimization iterations.

## E.2  DATASET STATISTICS

Table 10 summarizes the dataset sizes and evaluation metrics across domains.

Table 10: Dataset statistics and corresponding evaluation metrics.

| Domain | Dataset | #Train | #Test | Metric |
|---|---|---|---|---|
| Code Generation | HumanEval | 33 | 131 | pass@1 |
| | MBPP | 86 | 341 | pass@1 |
| Math Reasoning | GSM8K | 264 | 1055 | Accuracy |
| | MATH | 119 | 486 | Accuracy |
| Tool Used | GAIA | 94 | 372 | Accuracy |
| | ALFWorld | 27 | 107 | Success Ratio |
| Multi-hop QA | HotpotQA | 200 | 800 | F1 Score |

## F PROMPT EXAMPLES

**Example of workflow**

```python
class MultiAgentSystem:
    def __init__(self, name: str, tools=None) -> None:
        self.name = name
        self.tools = tools

    async def run(self, task: str):

        from Agent_async import Actor

        # Initialize agents
        actor = Actor(self.tools)

        # Run agent flow
        prompt = f"As a mathematician,
        solve the following complex math problem: {task}"
        res = await actor.process(prompt)

        return {"answer": res}
```

**X1: Diversity Exploration**

```python
prompt_content = (
    f"Create a novel multi-agent workflow for solving complex multi-
        step reasoning problems using LLMs.\n"
    f"I have {num_indiv} existing agent flows with their codes as
        follows:\n"
    f"{prompt_indiv}"
    "Please help me create a new agent flow that has a totally
        different form from the given ones in structure and prompts.\
        n"
    "Focus on generating a workflow with completely novel topology
        and agent interaction patterns, diverging from all structural
        templates of the existing flows.\n"
    f"{self.format_prompt}"
    f"{self.agent_blocks}"
    f"{self.prompt_law}"
    "First, describe your new agent flow and main steps in one
        sentence. "
    "Then, please return the Python implementation of the
        MultiAgentSystem class in JSON format. "
    "{'plan': str, 'code': str, 'num_agent': int}"
)
```

**X2: Conceptual Exploration**

```python
prompt_content = (
    f"Design a new multi-agent system for problem solving using LLMs
        by reimagining core functions from existing flows.\n"
    f"I have {num_indiv} existing agent flows with their codes as
        follows:\n"
    f"{prompt_indiv}"
    "Please help me create a new agent flow that keeps the high-level
        intent or task decomposition ideas of the old ones, but
```

```
        introduces different agent types, roles, and data flow
        structures.\n"
    f"{self.format_prompt}"
    f"{self.agent_blocks}"
    f"{self.prompt_law}"
    "First, describe your new agent flow and main steps in one
        sentence. "
    "Then, please return the Python implementation of the
        MultiAgentSystem class in JSON format. "
    "{'plan': str, 'code': str, 'num_agent': int}"
)
```

**Y1: Fine Optimization**

```
prompt_content = (
    f"Improve an existing high-performance agent workflow through
        detailed refinements for efficiency and clarity.\n"
    f"I have {num_indiv} existing agent flows with their codes as
        follows:\n"
    f"{prompt_indiv}"
    "Please generate a new agent flow that improves on the most
        effective existing one by optimizing prompts, reducing
        redundant links, and increasing information clarity, while
        preserving the overall structure.\n"
    f"{self.format_prompt}"
    f"{self.agent_blocks}"
    f"{self.prompt_law}"
    "First, describe your new agent flow and main steps in one
        sentence. "
    "Then, please return the Python implementation of the
        MultiAgentSystem class in JSON format. "
    "{'plan': str, 'code': str, 'num_agent': int}"
)
```

**Y3: Agent Role Specialization**

```
prompt_content = (
    f"Design a new multi-agent system that enhances role
        specialization to increase efficiency and collaboration.\n"
    f"I have {num_indiv} existing agent flows with their codes as
        follows:\n"
    f"{prompt_indiv}"
    "Please help me generate a novel agent flow where each agent has
        clearly defined specialized roles and improved collaboration
        patterns, optimizing division of labor and communication
        efficiency.\n"
    f"{self.format_prompt}"
    f"{self.agent_blocks}"
    f"{self.prompt_law}"
    "First, describe your new agent flow and main steps in one
        sentence. "
    "Then, please return the Python implementation of the
        MultiAgentSystem class in JSON format. "
    "{'plan': str, 'code': str, 'num_agent': int}"
)
```

