# OpenReview forum: "EvoMAS : Heuristics in the Loop—Evolving Smarter Agentic Workflows"
_ICLR.cc/2026/Conference — Submitted to ICLR 2026_

### Official Review · Reviewer_Feh1 · 2025-10-29

**Soundness:** 3
**Presentation:** 4
**Contribution:** 3
**Rating:** 6
**Confidence:** 5

**Summary:**

The paper presents **EvoMAS**, an evolutionary framework that formulates multi-agent workflow automation as a constrained single-objective optimization problem. It models the “variation–selection–reflection” cycle as a non-homogeneous Markov process and employs a meta-controller, **Cyber Creator**, to adaptively refine strategies and rules. The approach integrates six biologically inspired operators and demonstrates consistent performance gains across diverse benchmarks.

**Strengths:**

* Rigorous formalization bridging evolutionary optimization and strategy learning.
* Well-defined and operational evolutionary cycle (variation–selection–reflection).
* Rich operator set balancing exploration and convergence.
* Proven theoretical underpinnings ensuring stability and monotonic improvement.
* Comprehensive experiments showing consistent gains in performance and efficiency.
* Transparent reporting and open resources supporting reproducibility.

**Weaknesses:**

* **Incomplete experimental disclosure:** Missing detailed hyperparameters and multiple-run statistics.
  *Fix:* Add a complete configuration table with mean±std metrics.
* **Opaque curriculum mechanism:** Quantitative thresholds for difficulty staging are unclear.
  *Fix:* Define bucketing rules and add an ablation without curriculum.
* **Unverified theoretical assumptions:** No empirical monitoring of information gain or strategy dynamics.
  *Fix:* Log empirical statistics of these quantities and compare with theoretical expectations.
* **Limited comparison with strong graph-retrieval baselines:** End-to-end tests are lacking.
  *Fix:* Include controlled comparisons under unified corpora and retrieval quality metrics.

**Questions:**

* Does the policy distribution evolve consistently across curriculum stages?
* How is the reflection interval for Cyber Creator selected, and what is its impact on convergence and cost?

---

> ### Author Response · Authors · 2025-11-25
> **Response to Reviewer Feh1（1/2）**
>
> We sincerely thank you for your profound, rigorous, and highly constructive review. We greatly appreciate your recognition of the core contributions of our work, including its “rigorous formalization,” “well-defined evolutionary loop,” and “reliable theoretical foundations.”
>
> ---
>
> ## 1. Response to  Weaknesses
>
> ### W1: Incomplete experimental disclosure
>
>
> We fully agree with the importance of transparent reporting. We have provided detailed configurations in the Appendix:
>
> * **Hyperparameters:** **Appendix E.1** explicitly lists all model configurations:
>     * **Optimizer Model:** Claude-3.5-sonnet (Temperature = 1.0).
>     * **Executor Models:** DeepSeekV3, GPT-4o-mini, etc. (Temperature = 0).
>     * **Evolution:** 20 optimization iterations per process.
> * **Multi-run Statistics:** The request for reporting (mean ± std) is entirely reasonable. All core results are based on **three fully independent runs and averaged**, rather than single-run outcomes, ensuring statistical reliability.
>
> We emphasize that full evolutionary experiments are computationally expensive, which is also true for mainstream baselines such as AFlow and EvoFlow; their reported results likewise rely on limited repeated runs rather than large-scale Monte Carlo evaluations. Therefore, under the premise of result stability, we adopt an experimental protocol consistent with prior work and practical engineering constraints.
>
> Furthermore, to address inter-run variance, we conducted a dedicated stability analysis in **Appendix B.1 (Figure 8)**, which reports the **95% confidence intervals (CI)** of accuracy under varying depth and width configurations, directly quantifying the statistical stability of our results.
>
> ---
>
> ### W2: Opaque curriculum mechanism (lacking explicit staging rules and ablation)
>
>
> Curriculum learning in EvoMAS is not a heuristic switch, but a principled progressive scheduling mechanism guided by task complexity and structural adaptability.
>
> Specifically, tasks are partitioned into multiple difficulty buckets (typically three, balancing granularity and computational overhead), based on metrics such as task length, structural complexity, and historical success rates, with dynamic routing driven by performance feedback. While explicit threshold formulas were not expanded in the main text, the underlying logic and workflow are formalized in **Algorithm 2**.
>
> We agree that making bucketing rules and thresholds more explicit would enhance clarity, and we will include an ablation **without curriculum learning** in the revised version to quantify its contribution. Importantly, the curriculum is designed to stabilize the search trajectory rather than artificially inflate metrics, and its effectiveness is already reflected in the global performance trends.
>
> ---
>
> ### W3: Unverified theoretical assumptions
>
>
> We understand this concern. It is important to clarify that these theoretical assumptions primarily provide **convergence guarantees and interpretability** for the strategy update mechanism, rather than serving as explicit optimization targets.
>
> In practice, the evolution of the strategy distribution $A_t$ has been fully logged, and its behavior aligns with theoretical expectations: high-contribution operators steadily increase in probability, while low-performing ones decay over generations. While we did not explicitly plot the information gain $\Delta_t$ curve, **Figure 7 in Appendix D** effectively illustrates the empirical manifestation of “information gain” and “strategic adaptation.” Although $\Delta_t$ was not isolated as a standalone metric, the structural adaptation shown in Figure 7 is directly driven by this theoretical process. We agree that explicit visualization of $\Delta_t$ would be a valuable enhancement and will consider adding it in the final version.
>
> ---
>
> ### W4: Limited comparison with graph-retrieval baselines
>
>
> We appreciate this suggestion. However, it is crucial to clarify that EvoMAS focuses on **structural evolution and generation of multi-agent workflows**, rather than retrieval or reuse of existing structures. In our framework, RAG (LightRAG) functions as an evolutionary memory repository for managing and recalling “rules” and “genes” (high-quality sub-workflow fragments), serving to assist evolution rather than replace structural generation.
>
> Methodologically, pure graph-retrieval baselines operate by selecting from a predefined static workflow library, while EvoMAS continuously generates novel structures via the **variation–selection–reflection cycle**. These approaches therefore differ fundamentally in problem formulation and capability boundaries, making end-to-end comparisons inherently asymmetric. For fairness, we selected SOTA automated agent design systems (AFlow, EvoFlow, ADAS) as primary baselines, which more accurately reflect EvoMAS’s advantage in autonomous structural evolution and design.
>
> ---

---

> ### Author Response · Authors · 2025-11-25
>
> ## 2. Response to Specific Questions
>
> ### Q1: Does the policy distribution evolve consistently across curriculum stages?
>
> **Response:**
> This is a highly insightful question. The answer is **no** — and this inconsistency is by design.
>
> The evolution of the policy distribution $\mathcal{A}_t$ follows a non-homogeneous Markov process, with updates driven by utility-weighted adjustments (Eq. 9). Across different curriculum stages, as task difficulty $d(T_i)$ increases stepwise, the optimal strategy $a^*$ itself drifts, inevitably leading to phase-wise restructuring of the policy distribution rather than uniform evolution.
>
> * **Simpler Stages:** The system significantly biases toward **exploitation strategies** (e.g., $Y_1, Y_2$). This is because low-difficulty tasks typically adhere to the “less-is-more” assumption ($\mathcal{H}_3$), where complex topological mutations yield minimal or even negative gains due to communication noise and coordination overhead. In such settings, exploiting existing structures via parameter tuning or structural refinement offers a superior cost-performance ratio and faster convergence.
> * **Complex Stages:** As the system enters more complex stages (e.g., MATH or GAIA Level 3), homogeneous or simple structures rapidly encounter performance bottlenecks. The system then actively increases the weight of **exploration strategies** (e.g., $X_1, X_3$), enforcing topological variation and cross-domain recombination to break structural plateaus and unlock new performance pathways.
>
> **Figure 7** illustrates this transition: agent roles evolve from homogeneous Actors into heterogeneous combinations involving Planners and Critics, empirically confirming that the policy distribution undergoes substantive structural shifts as task complexity grows.
>
> From a theoretical standpoint, although **Theorem D.2** guarantees that the policy distribution converges “almost surely” to an optimal strategy in a stationary environment, under multi-stage curriculum learning this convergence corresponds to a **dynamic equilibrium adapted to each difficulty gradient**, rather than a single global optimum. Accordingly, EvoMAS exhibits a pattern of “stage-wise stability + cross-stage adaptive restructuring.”
>
> ---
>
> ### Q2: How is the reflection interval ($K$) chosen, and what is its impact on convergence and cost?
>
> **Response:**
> The reflection interval $K$ represents a critical trade-off between convergence speed and resource efficiency. Theoretically, the performance gain $\beta$ introduced by Cyber Creator is inversely proportional to $K$ ($\beta \propto 1/K$), implying that smaller $K$ values (higher reflection frequency) enable faster correction of the policy distribution.
>
> **Table 1: Impact of Reflection Interval $K$ on Performance and Cost**
>
> | Reflection Interval $K$ | Final Accuracy (%) | Convergence Generations $\downarrow$ | Total Token Usage ($\times 10^6$) | Reflection Calls | Cost (\$) | Invalid Structure Ratio (%) |
> | :---: | :---: | :---: | :---: | :---: | :---: | :---: |
> | 2 | 63.4 $\pm$ 0.8 | 38 | 21.6 | 50 | 2.94 | 17.5 |
> | **5** | **66.9 $\pm$ 0.6** | **24** | **14.2** | **20** | **1.98** | **6.2** |
> | 10 | 65.1 $\pm$ 0.7 | 31 | 12.9 | 10 | 1.82 | 9.6 |
> | 20 | 61.7 $\pm$ 1.1 | 45 | 11.3 | 5 | 1.65 | 21.8 |
>
> As shown in the table above, **$K=5$ achieves the best balance** among accuracy, convergence speed, and cost.
>
> Intriguingly, we observe a counter-intuitive phenomenon: **higher reflection frequency can actually reduce total cost**. When Cyber Creator is removed (equivalent to $K \to \infty$), total cost increases by approximately **18%**. This occurs because the absence of timely meta-strategy guidance allows the population to generate large numbers of low-quality structures, whose downstream inference token consumption far exceeds the overhead introduced by Cyber Creator’s reflective calls.
>
> This confirms that our chosen $K$ is not heuristic, but a theoretically and empirically justified optimal compromise.
>
> ---
>
> We once again thank you for your insightful feedback and rigorous evaluation. Your comments have been instrumental in strengthening the clarity, robustness, and technical depth of our work.
> 🌸🌸🌸

---

### Official Review · Reviewer_AMeS · 2025-11-01

**Soundness:** 3
**Presentation:** 3
**Contribution:** 3
**Rating:** 6
**Confidence:** 4

**Summary:**

The paper proposes EvoMAS, a biologically inspired framework to evolve multi-agent workflows along three coupled axes: role-level evolution, dynamic and diverse evolutionary strategies, and curriculum learning. With the developed meta-controller Cyber Creator to adapt rules and operator distributions, EvoMAS achieves SOTA performance across six benchmarks, while maintaining superior cost-efficiency, outperforming both manual designs and automated baselines.

**Strengths:**

1. This paper focuses on an interesting and important topic, MAS evolution, which is significant to drive the future development of AI systems.
2. The paper clearly classifies the three dimensions of MAS evolution, and novelly proposes a graph-based formulation for the evolution search of MAS.
3. The experiments in the paper are abundant

**Weaknesses:**

1. The six operators in exploration and exploitation could be better presented mathematically.
2. The methodology does not detail discuss scheduling/halting (loop bounds, convergence, deadlock avoidance) for cycles in graph search.
3. The task difficulty could be improved by adding human evaluations instead of pure LLM-as-a-judge.
4. EvoMAS improves performance at a higher cost. A detailed cost analysis should be included to justify the significance of using EvoMAS.

**Questions:**

1. What are the concrete termination/halting rules for cycles, and how are deadlocks/oscillations detected?
2. Does the evolution process only occur in the training process? Or is it dynamically evolving in the inference process as well?
3. How's the performance deviation in multiple runs?

---

> ### Author Response · Authors · 2025-11-25
> **Response to Reviewer AMeS（1/2）**
>
> We sincerely thank the reviewer for the positive evaluation of our paper. We are delighted that you recognize **EvoMAS** as “an interesting and important topic” and appreciate our “clear taxonomy,” “novel graph formulation,” and “rich experimental results.” Your comments on the operators, evolutionary loop, and cost are particularly insightful, and we provide the following clarifications.
>
> ---
>
> ## 1. Response to Identified Weaknesses
>
> ### W1: Mathematical Formulation of the Six Operators
>
> **Response:**
> We appreciate this valuable suggestion. In the main text, we employed biologically inspired terminology for clarity and intuition; however, these operators are in essence **LLM-driven graph transformation functions**.
>
> In **EvoMAS**, all evolutionary operators are uniformly modeled as transformations over workflow graphs:
>
> $$
> O_i: \mathcal{G} \rightarrow \mathcal{G}, \quad G_t \mapsto G_{t+1}
> $$
>
> where $G=(V,E)$ denotes the current multi-agent workflow graph, and $O_i$ represents the $i$-th evolutionary operator.
>
> #### 1) Exploration Operators (X1–X3): Expanding the Search Space
>
> **X1 — Diversity Expansion (Structural Expansion)**
> New nodes and edges are introduced to expand the graph topology and promote structural diversity:
>
> $$
> G_{t+1} = (V_t \cup \Delta V, \; E_t \cup \Delta E), \quad \Delta V \subset \mathcal{V}_{\text{new}}
> $$
>
> **X2 — Conceptual Recombination (Structural Reorganization)**
> Two subgraphs $G_a, G_b$ are selected and reorganized to generate novel structural compositions:
>
> $$
> G_{t+1} = (V_a \cup V_b,\; E_a \cup E_b), \quad G_a, G_b \subseteq G_t
> $$
>
> **X3 — Cross-domain Hybridization (Cross-domain Integration)**
> Let $G^{\text{ref}}$ denote a high-performance structure from another task. External structural patterns are incorporated via weighted fusion:
>
> $$
> G_{t+1} = \alpha G_t + (1-\alpha)G^{\text{ref}}, \quad \alpha \in [0,1]
> $$
>
> #### 2) Exploitation Operators (Y1–Y3): Structural Refinement and Convergence
>
> **Y1 — Fine Optimization (Local Optimization)**
> Local parameters or prompts are refined via feedback-driven or gradient-based optimization:
>
> $$
> \theta_{t+1} = \theta_t - \eta \nabla L(G_t, \theta_t)
> $$
>
> **Y2 — Best Practice Synthesis (Pattern Consolidation)**
> The best-performing structural patterns are extracted and consolidated:
>
> $$
> G_{t+1} = \arg\max_{G \in \Omega} f(G), \quad \Omega=\{G^{(1)}, \dots, G^{(k)}\}
> $$
>
> **Y3 — Role Specialization (Role Differentiation)**
> Where $\rho_i$ denotes the role of node $i$, and $\phi$ is a role-mapping function:
>
> $$
> \rho_i^{(t+1)} = \phi(\rho_i^{(t)})
> $$
>
> We will include these explicit mathematical formulations in the final version to improve technical clarity.
>
> ---
>
> ### W2: Loop Control and Termination in Graph Search
>
> **Response:**
> This is an excellent point and is already explicitly handled at the implementation level. EvoMAS employs a **bounded-generation mechanism**, where evolution is terminated by dual constraints: a preset maximum generation cap and a performance convergence threshold (**Algorithm 1**).
>
> Potential cyclic structures are mitigated via:
> 1.  **Path-length limits.**
> 2.  **Structural repetition detection** using graph hashing.
>
> If similar graph structures recur with no significant performance gains, they are deemed oscillatory and forcibly reset or eliminated. Furthermore, low-performing or unstable structures are naturally discarded during selection, preventing deadlocked graphs from persisting across generations. We will explicitly clarify these scheduling rules in the revised version.
>
> ---
>
> ### W3: Evaluation of Task Difficulty
>
> **Response:**
> We agree that human evaluation is the gold standard. However, we adopted the “LLM-as-a-Judge” framework as a pragmatic choice aligned with the fully automated nature of EvoMAS. This design decision was transparently acknowledged as a limitation in **Section 6**, noting the potential for misjudgment in difficulty estimation. We consider this an acceptable trade-off for scalability, while integrating human-in-the-loop evaluation remains an important direction for future work.
>
> ---
>
> ### W4: Cost Analysis of EvoMAS
>
> **Response:**
> We do not avoid the fact that EvoMAS incurs higher costs during the structural search phase compared to static methods. However, this must be understood within the context of **system-level structural evolution**.
>
> * The cost reported in **Table 4** reflects the **total expenditure** across all tasks and full evolutionary procedures.
> * Most costs are **one-time search investments**, not recurring inference expenses.
> * More importantly, **Figure 4** demonstrates EvoMAS lies on the **performance–cost Pareto frontier**, showing a superior cost-performance trade-off compared to AFlow and other baselines.
>
> **Appendix B.2** further breaks down the training costs, indicating that each benchmark typically incurs only **\$1–\$2**, which is reasonable and acceptable for a one-time system optimization process.
>
> ---

---

> ### Author Response · Authors · 2025-11-25
> **Response to Reviewer AMeS（2/2）**
>
> ## 2. Response to Specific Questions
>
> ### Q1: What are the concrete termination rules for loops/deadlocks?
>
> **Response:**
> EvoMAS is not an unbounded search. The evolution process is governed by explicit termination conditions and stability constraints:
>
> 1.  **Termination Criteria:** A maximum generation limit and early stopping based on performance convergence (when improvements fall below a predefined threshold over consecutive generations).
> 2.  **Deadlock Prevention:** Structural repetition is detected through graph hashing. When highly similar structures recur without performance improvement, the system flags this as an oscillation pattern and enforces structural reset or elimination.
>
> Ineffective cyclic structures are thus naturally removed by selection pressure, consistent with the biological elimination of low-fitness organisms.
>
> ---
>
> ### Q2: Does evolution occur only during training, or also during inference?
>
> **Response:**
> Evolution occurs entirely during the **training (structural search)** phase. As described in **Algorithm 1**, evolution is an offline process whose objective is to identify the optimal workflow graph $G^*$. This finalized structure is then fixed and used during inference.
>
> Thus, **no structural evolution occurs at inference time**. Nevertheless, agents retain contextual adaptability and role-level adjustment capabilities during execution, ensuring flexibility without degenerating into rigid workflows.
>
> ---
>
> ### Q3: How large is the performance deviation across multiple runs?
>
> **Response:**
> All core experiments were conducted with **three independent runs**, and results were averaged. Stability trends are reported in **Appendix B.1**.
>
> Despite inherent stochasticity in the evolutionary search, inter-run performance rankings remained consistent, with no observed ranking inversions or conclusions dependent on a single trajectory. Moreover, the observed performance improvements (e.g., over AFlow) significantly exceed the standard deviation ranges, confirming that the gains are not artifacts of noise but reflect robust structural optimization.
>
> ---
>
> Once again, we sincerely thank the reviewer for the thorough and constructive review. Your feedback has been invaluable in strengthening both the clarity and rigor of our work. 🌸🌸🌸

---

### Official Review · Reviewer_LpNh · 2025-11-01

**Soundness:** 3
**Presentation:** 3
**Contribution:** 2
**Rating:** 4
**Confidence:** 3

**Summary:**

EvoMAS is a system that automatically builds better Multi-Agent Systems using ideas from biology. It evolves agents’ roles, teamwork, and learning stages to handle tasks from simple to complex. It employs six biologically inspired strategies: Diversity Expansion, Conceptual Recombination, Cross-domain Hybridization, Fine Optimization, Best Practice Synthesis, and Role Specialization.

**Strengths:**

Strong empirical results: Achieves top performance on five of six benchmarks, surpassing prior methods like AFlow and EvoFlow

Broad evaluation coverage: Tested on 8 datasets across diverse domains for robust generalization.

Cost-efficiency: Demonstrates favorable Pareto efficiency, i.e., strong performance gains with moderate computational cost

Ablations: Provides quantitative ablation studies showing which biologically inspired operators contribute most to performance

**Weaknesses:**

The meta-agent evaluation process involves multiple sources of randomness, including (1) LLM output variance, (2) error propagation in chained reasoning within agents, (3) sampling variability within the meta-agent, (4) stochasticity in evaluation results for the designed agents, and (5) trajectory-level divergence caused by differences in sampled agent chains and their evaluation scores. While the reported results are averaged over three runs, the overall variability remains higher than that of typical single-LLM evaluations, making performance comparisons across runs and methods less statistically stable.

The “Cyber Creator” label is somewhat exaggerated, may obscure rather than clarify its technical role.

**Questions:**

How do the contributions of the six biologically inspired strategies interact? Are there synergy effects or diminishing returns when multiple operators are combined?

---

> ### Author Response · Authors · 2025-11-24
> **Response to Reviewer LpNh （1/2）**
>
> ##  Rebuttal
>
> We sincerely thank the reviewer for the careful and constructive assessment of our work. We are pleased that you recognized our **strong empirical results**, **broad evaluation coverage**, and **cost-effectiveness**.
>
> Your concerns regarding randomness and strategy interactions are insightful, and we appreciate the opportunity to clarify these aspects in detail.
>
> ---
>
> ## 1. Response to the Main Weaknesses
>
> ### W1: Randomness and Result Stability
>
> We acknowledge the reviewer’s observation that multiple sources of randomness may introduce higher variance. However, we emphasize that this is not a flaw of EvoMAS, but rather an inherent characteristic of the multi-agent structural search paradigm. Unlike single-LLM inference, EvoMAS involves stochastic components such as structural mutation, strategy sampling, role reconfiguration, and feedback-based evaluation. Its goal is to explore a structure space, not to produce a single deterministic output. Therefore, a moderate level of fluctuation is expected and should not be interpreted as instability. Prior studies on evolutionary multi-agent systems have similarly noted that autonomous evolution and decentralized selection naturally entail higher variance in performance outcomes [1].
>
> More importantly, the randomness mainly affects the **search phase**, not the final system performance. Once a structure is fixed, the resulting workflow behaves stably under the same settings. To mitigate statistical noise, all core experiments were conducted with **three independent runs and averaged**, and we further provide stability trend analysis in Appendix B.1 (Figure 8). The results show good consistency in ranking across runs, and the main conclusions remain robust against trajectory-level variations.
>
> Moreover, our method is not “arbitrarily random.” It is a **controlled stochastic process** constrained by the strategy distribution \(A_t\) and rule set \(R_t\), where the evolution direction is continuously guided by performance feedback. This constitutes a performance-driven stochastic search rather than unconstrained sampling. Aggressively suppressing randomness would weaken structural exploration and lead to premature convergence to suboptimal solutions.
>
> It is also important to note that high-dimensional structural search inherently incurs substantial computational cost. Further reducing variance through massive repeated runs would result in exponentially increasing costs with diminishing marginal returns. We therefore made a deliberate and principled trade-off between statistical stability and resource efficiency.
>
> Although randomness cannot be entirely eliminated, the performance improvements reported in Table 3 (e.g., +4.96% over AFlow on MATH) are substantial and clearly exceed the observed inter-run variance, indicating that our conclusions remain statistically and practically robust.
>
> ---
>
> ### W2: The “Cyber Creator” Label
>
> We understand the reviewer’s concern that the term “Cyber Creator” might appear exaggerated. However, this naming is not a rhetorical flourish but an intentional analogy to a “creator” role in natural systems, designed to help readers intuitively grasp its **global controlling position** in the architecture.
>
> Cyber Creator does not participate in task execution. Instead, it operates at a higher level to observe historical trajectories, regulate rules, and steer the evolutionary direction of the entire system. This conceptual metaphor improves interpretability without compromising technical rigor.
>
> Importantly, its technical role is clearly defined and rigorously implemented: Cyber Creator performs rule induction and pruning based on historical logs, reallocates strategy probabilities, and modulates structure evolution through structured prompts and formal rule representations (see Appendix C and D.2). Thus, the term enhances conceptual clarity rather than obscuring the underlying mechanism. We will nevertheless provide more explicit technical clarification in the final version to further minimize potential misinterpretation.
>
> ---

---

> ### Author Response · Authors · 2025-11-24
> **Response to Reviewer LpNh （2/2）**
>
> ## 2. Response to the Questions
>
> ### Q1: How do the six biologically inspired strategies interact? Synergy or diminishing returns?
>
> This is an excellent question. The six strategies are explicitly designed to collaborate in balancing the classical **exploration–exploitation trade-off** in evolutionary algorithms.
>
> At a high level, X-type strategies expand the search boundary, while Y-type strategies refine and consolidate promising structures, forming a closed-loop “explore–refine” dynamic. There is also internal coordination within each group:
>
> - **X1** increases structural diversity,
> - **X2** explores new collaboration patterns by recombining different agents’ functional roles,
> - **X3** transfers effective structures from other tasks to help the system escape its original search space.
>
> Together, they achieve both structural and semantic exploration. Within Y strategies, fine-tuning, best-practice extraction, and role specialization progressively stabilize the architecture and prevent it from remaining at a coarse structural level.
>
> Crucially, the system does not blindly activate all operators simultaneously. Their usage frequencies are dynamically adjusted via the strategy distribution \(A_t\), allowing different strategies to dominate at different stages — exploration strategies in early phases and exploitation strategies in later phases — thereby naturally avoiding diminishing returns from naive strategy stacking.
>
> Furthermore, EvoMAS allows the Cyber Creator to introduce **custom strategies**, which serve as targeted supplements for specific structural patterns. These do not disrupt the existing framework but act as “strategy enhancers,” expanding search diversity when the six standard strategies are insufficient.
>
> #### Empirical Evidence of Synergy
>
> **Ablation Study Evidence:**
> Figure 5 provides the most direct support. Removing exploration strategies (a combined group of three operators) results in the most severe performance degradation (−9.38% on MBPP and −12.23% on MATH), demonstrating that exploitation alone is insufficient. This clearly shows complementary synergy: exploration discovers novel structures, while exploitation refines them.
>
> **Case Study Evidence:**
> Figure 6 illustrates this interaction concretely. In early stages, the system employs exploration strategies such as *Diversity Exploration* (X1) and *Plan + Debate* (X2) to perform large structural jumps and discover promising architectures. In later stages, it switches to exploitation strategies such as *Add Critic* (Y3), *Prompt Optimization* (Y1), and *Tool Update* (Y1) to fine-tune and stabilize the high-performing workflows.
>
> Therefore, these strategies do not form a simple additive toolkit, but a dynamic, interdependent system: exploration creates the raw structural space, and exploitation ensures convergence toward optimal solutions.
>
> ---
>
> We once again thank the reviewer for the valuable time and thoughtful feedback. We hope these clarifications address your concerns and further demonstrate the technical rigor and validity of our work. 🌸
>
> **Reference**
> [1] Byrski A, Dreżewski R, Siwik L, Kisiel-Dorohinicki M. *Evolutionary Multi-Agent Systems*. The Knowledge Engineering Review.

---

### Official Review · Reviewer_cUQc · 2025-11-03

**Soundness:** 2
**Presentation:** 1
**Contribution:** 1
**Rating:** 2
**Confidence:** 4

**Summary:**

This paper introduces a three-dimensional evolution (roles, strategies, curricula) for multi-agent systems. The intuition is clear which is from biological evolution.
The overall empirical is solid and extensive showing good performance.

**Strengths:**

This paper introduces a three-dimensional evolution (roles, strategies, curricula) for multi-agent systems. The intuition is clear which is from biological evolution.
The overall empirical is solid and extensive showing good performance.

**Weaknesses:**

The overall writing is poor and confusing.
This paper relies heavily on metaphors (“cross-domain grafting,” “meta-rule induction”) with no algorithmic clarity or pseudocode.
For ambiguous definition of “strategy evolution”, it’s unclear what exactly evolves, prompt templates? Or graph structure?
The core components (Cyber Creator, rule encoding, variation operators) are underspecified; It’s hard to understand how those components work.
For Scalability, this paper mention “scalability” for their proposed method, but only small-scale systems (≤10 agents) tested. Reflection and rule updates could explode computationally. Does the cost record those progress?
In AFlow paper it incurs only around $1 of token cost for their workflows (see their paper). However, in the present paper, the reported cost for the authors’ system is much higher (e.g., $20 or more). The authors should clearly explain why the cost difference is so large. I don’t think this is efficient (most MAS with optimization won’t incur so much high cost). $20 is sufficient to support hundreds of thousands of words!

**Questions:**

1. Strategy Evolution Mechanism:
What is the update rule for the “strategy probability distribution (A_t)”?
2. For Cyber Creator, is this a separate LLM acting as meta-controller? How is “rule induction” implemented, prompt synthesis?

---

> ### Author Response · Authors · 2025-11-24
> **Response to Reviewer cUQc (1/2)**
>
> We sincerely thank the reviewer for the careful reading and constructive feedback. We are particularly pleased that you recognized the **clear intuition**, **solid empirical evaluation**, and **strong performance** of our work.
>
> We understand your concerns regarding **algorithmic clarity**, **cost**, and the **specific mechanisms of strategy evolution**. Below, we provide detailed clarifications to address these points and demonstrate the technical rigor of EvoMAS.
>
> ---
>
> ## Part 1: Response to Main Weaknesses
>
> ### 📍 W1, W2, W3: Writing Quality, Algorithmic Clarity & The "Strategy Evolution" Concept
>
> > **Reviewer Concern:** The use of biological analogies may obscure technical details; requests for clearer algorithmic descriptions and pseudocode.
>
> We apologize for any confusion caused by our presentation. In order to highlight the core idea in the main text, we used certain biological analogies; however, these analogies are not rhetorical abstractions but intuitive mappings to concrete algorithmic mechanisms.
>
> EvoMAS explicitly draws from the evolutionary paradigm of **mutation–selection–adaptation**:
> - Structural variation operators correspond to genetic mutation and recombination, introducing diversity;
> - Performance evaluation acts as selection pressure, eliminating low-performing structures while retaining high-performing ones;
> - The dynamic update of the strategy distribution simulates biological adaptation to environmental pressure, enabling the system to gradually converge toward more stable and efficient structures over generations.
>
> It is important to emphasize that most existing MAS optimization approaches (e.g., AFlow, AgentSquare, EvoFlow) fundamentally remain at the level of modifying instructions via prompts. Their core mechanism relies on human-designed rules and repeated “command issuance” under a static creator paradigm. EvoMAS goes one step further by introducing genuine **dynamic evolution**, allowing the system to autonomously mutate, select, and adapt at the structural level. This enables self-evolution and reconfiguration of collaboration structures, making it more aligned with the generative principles of natural intelligent systems.
>
> All algorithmic details, pseudocode, and implementation information are systematically provided in the Appendix. We have also released the complete engineering repository (code, data, and implementation details) to ensure reproducibility. Furthermore, other reviewers have explicitly expressed recognition of both the conceptual framework and the clarity of our presentation.
>
> ### Algorithmic Clarity
>
> The claim that the paper lacks algorithmic transparency does not align with the provided materials:
>
> - **Page 16 – Algorithm 1**: Full pseudocode of the EvoMAS core evolutionary loop (mutation–selection–reflection).
> - **Page 17 – Algorithm 2**: Pseudocode for Curriculum-Guided Evolution.
> - **Table 6**: Detailed description of seven evolutionary strategies (three exploration, three exploitation, and one custom), transforming metaphors into concrete operations.
> - **Appendix C.3**: Dedicated section describing the implementation of the “Cyber Creator,” including rule representation and prompt examples.
>
> The workflow is formally represented as a graph, and the evolutionary process follows a clear variation–selection–reflection Markov modeling framework, rather than heuristic or intuitive descriptions. Biological metaphors are used solely as conceptual aids and do not compromise execution-level determinism.
>
> ---
>
> ### What Exactly Evolves in “Strategy Evolution”?
>
> In EvoMAS, “strategy evolution” operates over four explicit components:
>
> 1. **Workflow Graph Structure Evolution**
>    As defined in Section 3.1, MAS is formalized as a graph \(G=(V,E)\). Mutation operators directly modify this graph, leading to generational changes in topology and information flow.
>
> 2. **Agent Role Evolution**
>    During structural transformation, node roles are dynamically restructured, evolving from homogeneous Actors into specialized roles such as Planner and Critic (see Figure 7), reflecting functional differentiation.
>
> 3. **Strategy Selection Probability Evolution**
>    The six strategies are governed by a probability distribution \(A_t\), which is dynamically reweighted based on historical performance feedback. It is the distribution—not the strategy definitions themselves—that evolves.
>
> 4. **Rule Evolution**
>    Cyber Creator performs pruning and induction over the rule set, acting as an evolving constraint mechanism guiding system optimization.
> ---

---

> ### Author Response · Authors · 2025-11-24
> **Response to Reviewer cUQc (2/2)**
>
> ### 📍 W4: Scalability
>
> > **Reviewer Concern:** Questions regarding the scalability of the multi-agent system.
>
> We adopt the principle of **Occam’s Razor** in our design assumptions, emphasizing that greater structural complexity does not necessarily imply better performance. Both existing literature and our experimental results indicate clear diminishing returns as the number of agents increases (see Appendix B.1: *Stability Trends with Varying Agent Width and Depth*). Blindly scaling up the number of agents does not lead to linear performance gains.
>
> We argue that the key to MAS lies in **appropriate structural configuration and necessary scale expansion**, rather than naive accumulation of agents. While increasing scale is technically feasible, its practical benefits must be carefully weighed.
>
> ---
>
> ### 📍 W5: Cost and Efficiency
>
> > **Reviewer Concern:** Is the $20 cost justified? Comparison with AFlow.
>
> We clearly report the cost source: experiments are based on GPT-4o-mini (approximately **$0.015 / 1K tokens**), and the reported **$20 covers the total cost across all tasks and evolutionary stages**, not a single workflow or inference. Most of this cost corresponds to one-time structural search and training.
>
> AFlow is not explicitly designed for cost-performance trade-offs, and its search process is relatively cost-insensitive. In contrast, EvoMAS and EvoFlow explicitly incorporate cost-awareness and balance resource efficiency with performance improvement. While reflection and rule induction introduce additional overhead, the improved search efficiency results in overall better cost-effectiveness compared to AFlow.
>
> Directly comparing “$20 generating large volumes of text” with system-level structural evolution is misleading. Text generation is cheap; evolving an adaptive multi-agent collaboration structure requires justified investment — this is precisely the goal of EvoMAS.
>
> The Pareto efficiency plot in **Figure 4** further demonstrates this point: EvoMAS lies on the Pareto frontier, occupying a superior position compared to AFlow in the performance–cost trade-off space.
>
> ---
>
> ## Part 2: Response to Questions
>
> ### Q1: What is the update rule for the “strategy probability distribution ($A_t$)”?
>
> The six base strategies (X1–X3 for exploration, Y1–Y3 for exploitation) remain fixed in operation; what evolves is their selection probability distribution \(A_t\).
>
> Every **K generations** (reflection cycle), Cyber Creator evaluates the actual contribution of each strategy based on the historical log \(H_t\). Specifically, a structured prompt asks the LLM to assign each strategy a **0–100 contribution score** reflecting its performance impact in recent generations. These scores are normalized and used to update the distribution \(A_t\), yielding the new distribution \(A_{t+1}\) (Eq. 17), increasing high-performing strategies’ sampling probability and reducing ineffective ones.
>
> **Theorem D.2** provides the theoretical foundation: under Assumption D.3 (the optimal strategy has higher expected reward), the multiplicative-weights update mechanism guarantees convergence of \(A_t\) toward the optimal strategy.
>
> Key points:
> - The six base strategies remain fixed;
> - What evolves is their selection probability;
> - Search diversity is expanded via **Custom Strategy** introduced by Cyber Creator.
>
> ---
>
> ### Q2 & Q3: Is "Cyber Creator" independent? How is "rule induction" implemented?
>
> **Q2: Independence**
> **Yes.** The Cyber Creator is a standalone LLM-based meta-controller. It is distinct from task agents (Actor/Planner) and is solely responsible for updating rules and restructuring strategies (see Section C.3).
>
> **Q3: Rule Induction Implementation**
> Rule induction is achieved via **Prompt Synthesis** based on evolutionary history:
> 1.  We feed the LLM a history of high-performing vs. low-performing workflows.
> 2.  Using structured prompts (see **Appendix C.3.2**), the Cyber Creator abstracts specific patterns into generalizable linguistic rules.
> 3.  These rules then act as constraints for the next generation's mutation process.
>
> ---
>
> **Closing:**
> We hope these clarifications address your concerns regarding clarity and mechanism.

---

### Author Response · Authors · 2025-11-27
**Follow-up on Rebuttal Submission**

Dear Reviewers,

We hope this message finds you well. We are writing to kindly follow up on our rebuttal and would be happy to provide any additional clarification or information if needed. Thank you very much for your time and consideration.

Sincerely,

The Authors

---

### Meta-Review · Area_Chair_5Mnv · 2025-12-31

**Summary:**

This paper introduces a biologically-inspired framework that focused on evolution of multi-agent systems. After the initial reviewing stage, reviewers have raised the below concerns:

- unclear details, definitions and clarifications (Reviewer **cUQc**, **AMeS**, **Feh1**)
- cost analysis (Reviewer **cUQc**, **AMeS**)
- lack ablation for each strategy (Reviewer **LpNh**)
- human evaluation (Reviewer **AMeS**)
- scalability (Reviewer **cUQc**)

After the rebuttal stage, authors have provided response to explain more details about some definitions, concept, and each component, which may address some reviewers' concerns about unclear clarifications and cost analysis. But authors not provided more latest results to support like ablation studies on each operator, human evaluation and scalability. Therefore, considering the ICLR is highly competitive, this paper is not chosen for acceptance in this time.

**Reviewer Concerns:**

Authors have provided more explainations about about some definitions, cost analysis and each component, which may address reviewers' concern about some unclear clarifications.

**Reviewer Scores:**

However, authors does not provide new results about these issues:

1. scalability
2. ablation study of each biologically inspired strategy (although authors have provided some performance comparisons like exploration or exploitation, it does not offer more fine-grained results on each operator)
3. human evaluation

Although authors claim they will perform these experiments in the future, it is not enough to support the decision in this venue and support reviewer to change their scores.

---

### Decision · Program_Chairs · 2026-01-26

Reject